

# Development and field-testing of an online instrument for measuring the real-time oxidative potential of ambient particulate matter based on dithiothreitol assay

Joseph V. Puthussery, Chen Zhang, Vishal Verma

Department of Civil and Environmental Engineering, University of Illinois at Urbana-Champaign, Urbana,61801, USA

*Correspondence to*: Vishal Verma (vverma@illinois.edu)

**Abstract.** We developed an online instrument for measuring the oxidative potential (OP) of ambient particulate matter (PM) using dithiothreitol (DTT) assay. The instrument uses a mist chamber (MC) to continuously collect the ambient $PM_{2.5}$ in water,

followed by its DTT activity determination using an automated syringe pump system. The instrument was deployed at an urban site in the University of Illinois campus, and its field performance was evaluated by comparing the results with the offline DTT activity measurements of simultaneously collected PM-laden filters. The online DTT activity measurements correlated well with the offline measurements but were higher than both methanol (slope = 0.86, $R^2$ = 0.93) and Milli-Q water (slope = 0.52, $R^2$ = 0.86) extracts of the PM filters, indicating a better efficiency of MC for collecting the water-insoluble fraction of

PM. The hourly measurements of ambient $PM_{2.5}$ OP were obtained by running the online instrument intermittently for 50 days with minimal manual assistance. The daytime DTT activity levels were generally higher than at night. However, a four-fold increase in the hourly averaged activity was observed on the night of July 04 (Independence Day fireworks display). Diurnal profile of the hourly averaged OP during weekdays showed a bimodal trend, with a sharp peak in the morning (around 7:00 AM), followed by a broader afternoon peak, which plateaus around 2:00 PM, and starts subsiding at night (around 7:00 PM).

To investigate the association of the diurnal profile of DTT activity with the emission sources at the site, we collected time-segregated composite PM filter samples in four different time periods of the day [morning (7:00 AM - 10:00 AM), afternoon (10:00 AM - 3:00 PM), evening (3:00 PM – 7:00 PM) and night (7:00 PM – 7:00 AM)] and determined the diurnal variations in the redox active components [i.e. water soluble Cu, Fe, Mn, organic carbon, elemental carbon and water soluble organic carbon]. Based on this comparison, we attributed the daytime OP of ambient $PM_{2.5}$ to the water-soluble Cu from both exhaust

and non-exhaust emissions, whereas secondary particles formed by the photochemical transformation of primary emissions appear to enhance the OP of PM during the afternoon and evening period.

## 1 Introduction

Several recent studies have used the oxidative potential (OP) of ambient particulate matter (PM) as an indicator of the aerosol toxicity (Li et al., 2003; Knaapen et al., 2004; Steenhof et al., 2011; Bates et al., 2015; Tuet et al., 2016; Yang et al., 2016).

The underlying hypothesis in these studies is that ambient particles upon inhalation can catalyze the generation of reactive



oxygen species (ROS), creating a biochemical imbalance between oxidants and antioxidants, which lead to a state of the cellular oxidative stress (Knaapen et al., 2004). This inherent property of the ambient PM to induce oxidative stress is supposed to be more closely associated with the PM related adverse health effects (such as congestive heart failure, myocardial infarction, asthma etc.) than conventionally used PM mass concentrations (Bates et al., 2015; Maikawa et al., 2016;

Weichenthal et al., 2016; Yang et al., 2016; Abrams et al., 2017).

To quantify the OP associated with ambient PM, researchers have used both cell-based and cell-free assays. Cell based assays generally involve the use of fluorescent probes (such as 2′,7′-dichlorodihydrofluorescein diacetate, DCFH-DA) for measuring the PM induced intracellular ROS in alveolar macrophages or other cell lines (Landreman et al., 2008; Saffari et al., 2014; Tuet et al., 2016). These assays are laborious, complex and require long analysis time (Dungchai et al., 2013). Cell free assays,

on the other hand, use less controlled environment and provide faster estimation of OP (Fang et al., 2015). The dithiothreitol (DTT) assay is one of the most widely used cell free assay to measure the ambient particles' OP, which has shown good correlation with several biological markers of PM-induced adverse health effects (Steenhof et al., 2011; Delfino et al., 2013; Bates et al., 2015; Tuet et al., 2016; Yang et al., 2016; Abrams et al., 2017). In the DTT assay, PM catalyzes the transfer of electrons from DTT to oxygen, generating superoxide radicals. The consumption of DTT over time, which is reported as the

DTT activity or OP of the ambient PM, is assumed to be proportional to the concentration of redox active compounds in the PM sample (Kumagai et al., 2002; Cho et al., 2005).

Previous studies have suggested that OP of the ambient particles is affected by various factors such as, PM composition, size, and source (Li et al., 2003; Steenhof et al., 2011; Janssen et al., 2014; Tuet et al., 2016; Fang et al., 2017). Moreover, few studies have also indicated that the ambient particles' OP changes during the course of the day, due to influence of atmospheric

processing and aging (Verma et al., 2009; Saffari et al., 2014). The conventional approach for measuring the DTT activity of ambient PM involves long-duration sampling (typically 24-72 hours) using filters and extracting them in a suitable medium (e.g. water or methanol) (Verma et al., 2012; Fang et al., 2015). The laborious and time-consuming protocol of these methods limit their applications to only few samples, and thus make it almost impossible to fully capture the diurnal variations in the ambient particles' OP. Another issue in measuring OP of the ambient particles collected over filters is related with the chemical

stability of the PM samples. Generally, PM collected over filters might undergo chemical alteration during storage and extraction procedures (Majestic et al., 2007; Daher et al., 2011). Therefore, OP measured on the archived filters might not be truly representative of the real potential of ambient particles to generate ROS in a biological system after inhalation. For reducing these artefacts, certain researchers (e.g. Cho et al., 2005; Daher et al., 2011) used a liquid impinger (Biosampler, SKC West Inc.) to collect the concentrated slurry of ambient aerosols and measured its OP using the DTT assay. The direct collection

of particles into liquid helps to minimize the PM loss during sampling and extraction processes. However, this method is not fully automated, i.e. frequent manual observation is needed to maintain the optimum conditions for an upstream aerosol concentrator (Kim et al., 2001). Moreover, it requires relatively long sampling duration, i.e. at least six hours to collect enough volume of the concentrated slurry, making this approach unsuitable for obtaining a highly time-resolved OP data for the ambient particles.



Over the past decade, various online instruments have also been developed for measuring the ROS associated with ambient PM (Venkatachari and Hopke, 2008; Wang et al., 2011; Sameenoi et al., 2012; King and Weber, 2013; Wragg et al., 2016; Eiguren-Fernandez et al., 2017; Zhou et al., 2017). These instruments couple a direct aerosol into water collection device, e.g. particle-into-liquid sampler (PILS), liquid spot sampler (LSS) or mist chamber (MC) to an analytical system. However, most

of these instruments measure the particle-bound ROS, which represents only a small part of the particles' OP. To the best of our knowledge, only two online systems have been developed till now, which measure the OP based on DTT assay, i.e. Sameenoi et al., 2012 and Eiguren-Fernandez et al., 2017. Sameenoi et al. (2012) used a PILS coupled directly to a microfluidic electrochemical sensor for measuring the DTT activity of ambient particles. The online system performance was tested in the lab using standard reference dust and fly ash aerosolized samples. The measured DTT activity was reported to have a linear

correlation with the aerosol concentration. Eiguren-Fernandez et al. (2017) used an LSS for particle collection, followed by its DTT activity determination in a traditional protocol (i.e. light absorbance spectroscopy). However, a major shortcoming in these two studies was the lack or limited days of field measurements. The online system based on microfluidic electrochemical sensor (Sameenoi et al., 2012) was characterized only in the lab using simulated aerosol samples and has never been tested in the ambient environment (at least not reported in any publication). While, the system developed by Eiguren-Fernandez et al.

(2017) was tested in the field for only three days (with 3-hour time resolution), and the long-term stability of the system is yet to be evaluated. Moreover, the aerosol sampling devices, i.e. PILS (Sameenoi et al., 2012) and LSS (Eiguren-Fernandez et al., 2017) used for collecting the PM in these online systems are expensive (> USD 20,000).

Here, we discuss the development of a low-cost, automated online instrument to measure the hourly averaged OP of ambient $PM_{2.5}$ using DTT assay and its evaluation in the field conditions for over 50 days. A custom-built glass MC, which is

substantially cheaper (~USD150) than PILS or LSS was used for collecting the ambient PM suspension, which is then transferred to an automated analytical system for DTT activity determination. Our study reports, for the first time, the complete diurnal profile of the DTT activity of ambient $PM_{2.5}$ at an hourly resolution, obtained from 50 days of field deployment of the instrument. The diurnal variations in DTT activity were further compared with the diurnal variations in $PM_{2.5}$ chemical composition. The results presented in our study demonstrate the usefulness of obtaining a highly time-resolved data on $PM_{2.5}$

OP, to comprehend various emission sources and their relative importance in assessing the risk from $PM_{2.5}$ exposure over the course of the day.

## 2 Materials and Methods

### 2.1 Instrument set up

The basic layout of the instrument setup for measuring the $PM_{2.5}$ OP is shown in Fig.1. The entire system is housed in a 6'x

4'x 2' insulated aluminium chamber, built by the Civil and Environmental Engineering Machine Shop of the University of Illinois at Urbana-Champaign (UIUC). The temperature inside the chamber is maintained at 18°C using a portable air conditioner (Black and Decker BPACT14HWT). An MC was used for collecting the ambient $PM_{2.5}$ suspension in water. MCs



were originally designed for collecting the water-soluble gases with Henry's constant ($K_H$) > $10^3$ M/atm (Cofer III et al., 1985; Spaulding et al., 2002; Hennigan et al., 2009), however few studies (Anderson et al., 2008; King & Weber, 2013) have also used these devices for collecting ambient particles. The MC used in this study was custom built in the glassblowing shop at UIUC and operated at a flow rate of 42 LPM. Exact dimensions of the MC are given in Fig.S1 (supplementary information, SI). A 0.5" (OD) copper tube was used to connect inlet port of the MC to a $PM_{2.5}$ cyclone (University Research Glassware; URG, Carrboro, NC, USA). The working principle for particle collection in MC is explained in previous studies (Anderson et al., 2008; King and Weber, 2013). In brief, a predetermined volume of Milli-Q water is added, such that the water level remains above the bottom tip of the capillary tube inside the MC. As air flows through the tapered inlet, air pressure drops at the capillary nozzle tip, which causes the water in the bottom reservoir to rise through the capillary tube and create a fine mist inside the chamber. This mist impinges on a 47 mm TefSep polytetrafluoroethylene (PTFE) membrane hydrophobic filter (1μm pore size) (GVS, Life sciences, ME, USA) loaded in a filter pack (URG, Carrboro, NC, USA), attached at top of the MC. The mist continuously washes the particles as they are collected onto the filter, in addition to some direct scrubbing from the air stream. The airflow exiting the hydrophobic filter was continuously measured by an inline flow meter (10 – 100 SLPM, Dwyer Instruments, MI, USA). The airflow rate eventually starts decreasing with an increasing PM loading on the filter, i.e. gradual accumulation of some water-insoluble PM, which is not recovered in the MC suspension. Typically, the flow rate was found to decrease by approximately 10 % after 16 hours of continuous operation of the MC in our case. Therefore, the filter was changed twice in a day (at 7 AM and 7 PM) to ensure a near-constant flow rate of 42 LPM (± 2.5%). The online system couples the MC to an automated DTT activity determination system, consisting of two automated syringe pumps and a continuously shaken thermomixer (Fang et al., 2015; discussed in section 2.2) as shown in Fig.1.

## 2.2 Instrument operation and procedures

At the start of each sampling run, a fixed amount of Milli-Q water (which was varied depending on time of the day, discussed in section 3.2) was added to the MC using a programmable 10 mL syringe pump [Versa Pump 6 (V6); Kloehn Inc., Las Vegas, NV, USA]. The PM suspension (~1.75 mL) at the end of the sampling run (one hour) was withdrawn from MC and fed to the reaction vial (RV) of the automated analytical system for DTT activity measurement, using the same syringe pump. The DTT activity measurement system is based on the same protocol as described in Xiong et al. (2017) using two syringe pumps [V6 and Versa Pump 3 (V3); Kloehn Inc., Las Vegas, NV, USA), but slightly modified to handle small volume of the PM suspensions. The RV was placed in a thermomixer (550 rpm, Eppendorf North America, Inc., Hauppauge, NY, USA) maintained at a constant temperature of 37 °C, and continuously shaken. 0.25 mL DTT (1 mM solution, final concentration in the reaction vial = 100 μM) and 0.5 mL potassium phosphate buffer (pH 7.4; 0.5 μM) were then added to the RV, using a V3 and V6 syringe pump, respectively. To determine the DTT consumption rate, 50 μL aliquot from the RV was withdrawn by a V3 pump at fixed time points (3, 13, 23, 33 and 43 min), and mixed immediately with 0.5 mL DTNB (0.2 mM), added by the V6 pump in the measurement vial (MV). The residual DTT in the aliquot reacts with DTNB to form 2-nitro-5-thiobenzoic acid (TNB). TNB absorbs light sharply at a wavelength of 412 nm, and therefore, the absorbance was measured by pushing the





mixture (using the V6 pump) through a liquid waveguide capillary cell (LWCC-M-3100; World Precision Instruments, Inc., FL, USA) with an optical path length of 100 cm. The LWCC was coupled to an online spectrophotometer (Ocean Optics, Inc., Dunedin, FL, USA), which includes both ultraviolet-visible (UV−vis) light source (DH-Mini) and a multiwavelength light detector (Flame – S). The absorbance intensity at 600 nm was chosen as a reference (where there was no absorption from the

reaction mixture) to correct for the temporal drift in the absorbance spectrum. The absorbance at both wavelengths (412 and 600 nm), were continuously recorded and saved as a .CSV file using a data acquisition software (Spectra Suite, Ocean Optics, USA). Since, the absorbance is proportional to the quantity of DTT left in RV, the slope of the absorbance versus time (5-time points) gives the DTT consumption rate of the sample. The analytical part of the instrument along with MC were cleaned twice in a month by replacing all the chemicals first with methanol, followed by Milli-Q water, and running the automated system

script for a routine DTT assay protocol (five times with each solvent).

**2.3 Field deployment of the instrument**

The online instrument was deployed on the roof (height from the ground level ~ 25m) of a parking deck (north campus parking) located adjacent to a 4-lane street (University Avenue) in the university campus and operated intermittently for approximately 50 days (between May 31 and August 16, 2017). While analytical part of the instrument measured the DTT activity of a given

PM suspension, the MC simultaneously collected a new PM suspension. Thus, hourly measurements of ambient $PM_{2.5}$ OP were obtained with minimal manual assistance, i.e. only for replacing the filters (twice in a day), resetting the computer program script (once in a day), and refilling the DTT solution (once in a day) and other reagents as needed (typically after 4-5 days).

**2.4 Instrument blanks**

Two different types of instrument blanks were analyzed in this study: system blanks and method blanks. System blank was determined by filling Milli-Q water in MC, but without collecting any air sample, i.e. vacuum pump remained switched off. The water remained in MC for few (2-3) minutes after which its DTT activity was measured by the analytical system. Method blank (particle-free) was determined by connecting a HEPA filter upstream of MC and running it for one hour. Thus, measured DTT activity in the method blank would be due to possible contribution from water-soluble gases. System blanks were

collected everyday (i.e. 1 blank per day) while method blanks were collected 4 times in a week, during field deployment of the instrument (discussed in section 3.2). Thus, a total of 50 system blanks and 28 method blanks were collected throughout the sampling period.

**2.5 Instrument performance**

The instrument performance was assessed by calibrating analytical part (DTT activity determination) of the instrument using positive controls (both before field deployment and during operation), determining the limit of detection (LOD; from the blanks



data), and comparing the online versus offline results (see section 2.6). All the results pertaining to the instrument performance are discussed in the results sections (sections 3.1, 3.3 and 3.4).

## 2.6 Online versus offline system comparison

The DTT activity measurements from the online system were compared with a conventional filter sampling and extraction method followed by offline DTT activity analysis of the extracts. Ambient filters (N=36), each for a duration corresponding to the MC sampling (i.e. 60 minutes), were collected for offline analysis from the same site (roof of north campus parking) on different days (see Table S1 in SI), while the online instrument was also running continuously. 47 mm TefSep hydrophobic filters (same as used in the filter pack of the MC) were used for collecting $PM_{2.5}$, at a flow rate of 42 LPM. After each sampling run, the filters were brought back to the lab and immediately stored in a freezer (-20°C). All the filters were extracted and analyzed within 12 hours of sampling. Limited mass loadings on these filters didn't allow us to extract each filter in both water and methanol. Therefore, 20 filters were extracted in Milli-Q water (1.75 mL volume per filter), while the remaining 16 filters were extracted in methanol, by sonication (Cole-Parmer Ultrasonic Cleaner 8891, IL, USA) for 30 minutes. The methanol extracts were evaporated (to <100 µL) by blowing high-purity nitrogen into the vial. The concentrated extracts were then reconstituted by adding Milli-Q water (final volume = 1.75 mL). The samples were immediately analyzed for DTT activity by a semi-automated system developed in our lab (Xiong et al., 2017). Field blank filters (N = 6) were similarly collected, processed (i.e. 3 filters extracted in Milli-Q water and 3 in methanol) and analyzed for the DTT activity.

## 2.7 Time-segregated composite sampling and chemical analysis

To infer the associations between diurnal trend in the ambient $PM_{2.5}$ DTT activity and $PM_{2.5}$ chemical composition, time-segregated composite $PM_{2.5}$ samples were collected for 10 weekdays in August 2017 (the exact dates are provided in the SI; Table S1). The composite samples were collected onto both pre-baked 8"x10" quartz filters (Tissuquartz 2500QAT-UP, Pall life Sciences, Port Washington, NY; referred as QS thereafter) and 8"x10" teflon filters (Zefluor PTFE Membrane Filters, Pall Laboratory, NY; referred as TS thereafter) using two high volume samplers (HiVol, Tisch Environmental, nominal flow rate 1.13m³ min⁻¹, $PM_{2.5}$ impactor), at the same site. These composite samples were collected in four different time periods of the day: morning (7:00 AM - 10:00 AM), afternoon (10:00 AM - 3.00 PM), evening (3:00 PM – 7:00 PM) and night (7:00 PM – 7:00 AM), i.e. the filters were changed four times in a day, but the same set of filters were used for all ten days. Thus, a total of eight samples were collected: QS1 and TS1 (morning; total hours of sampling = 30 hours), QS2 and TS2 (afternoon; total hours of sampling = 50 hours), QS3 and TS3 (evening; total hours of sampling = 40 hours), and QS4 and TS4 (night; total hours of sampling = 120 hours). All samples were stored in a freezer at -20°C between sampling intervals. The total $PM_{2.5}$ mass loading on the filters was determined by weighing (both pre- and post-sampling) the filters using a lab scale digital balance (± 0.2 mg readability, A-120S, Sartorius, Gottingen, Germany). Prior to weighing, the filters were equilibrated for 24 hours at 20°C and 50% relative humidity (RH).



The concentrations of elemental carbon (EC) and organic carbon (OC) were measured on a small section (1 cm$^2$) of the quartz filters as per National Institute for Occupational Safety and Health (NIOSH) protocol, using a thermal/optical transmittance (TOT) analyzer (Sunset Laboratory) (Birch and Cary, 1996). For the rest of the chemical analysis, teflon filter punches (25 mm diameter) were extracted into Milli– Q water by sonication (30 minutes). The water-soluble organic carbon (WSOC)

concentration in these extracts was measured using a total organic carbon analyzer (TOC-VCPH, Shimadzu Co. Japan) (Wang et al., 2018). The concentrations of water-soluble Cu, Fe and Mn in the extracts were determined by inductively coupled plasma mass spectrometry (ICP-MS) (Perkin Elmer, Waltham, MA) (Yu et al., 2018).

## 3 Results and Discussion

### 3.1 Calibration of the analytical system for DTT activity measurement

Before field deployment, the analytical measurement part of the online DTT instrument was calibrated by measuring DTT activity of 9,10-phenanthraquinone (PQN). Here, sampling part of the instrument, i.e. MC, was replaced with a 14-port multi-position valve (VICI® Valco Instrument Co. Inc., USA) to consecutively select different PQN standard solutions of known concentrations for analysis. The linear plot of DTT consumption rate (nmol/min) versus PQN concentration had a slope of 7.55 ± 0.54, with a coefficient of determination (R$^2$) of 0.98 (Fig.S2 in SI). The slope of calibration is very close to the slope

obtained earlier by Fang et al. (2015) using a similar automated DTT system. During the field operation, the analytical part of the instrument was calibrated at least once in 15 days by following the same procedure, i.e. replacing the MC with the multi-position valve.

### 3.2 Effect of evaporation in MC on DTT activity of ambient PM

During air sampling, the volume of water inside MC reduces due to evaporative loss. There are two concerns associated with

this loss of water. First, if the water level drops below the capillary, the mist formation is stopped, and the filter will collect particles by dry sampling (i.e. without mist formation). The rate of evaporation is largely governed by the ambient relative humidity (RH), which changes diurnally. Therefore, to address this concern, the Kloehn control program script was modified to adjust the amount of water added to the MC at the beginning of each sampling run, based on the forecasted hourly RH at the site  (AccuWeather, 2018). Typically, the RH was highest (78 ± 6 %) during night and early morning, and lowest (50 ± 8

%) in the afternoon period. Table S2 in SI shows the volume of water added to the MC for various ranges of RH observed at the sampling location. Adjusting this amount of water ensured at least 55 minutes of wet sampling (i.e. sampling PM with simultaneous mist formation) for each collected PM suspension, with rest as the dry sampling. Note, a fraction of the particles collected during dry sampling is extracted in the next run of wet sampling, which could lead to a small bias in the PM mass carried over to RV from the actual PM mass corresponding to a sampling run. However, this bias is not expected to be

significant given a very low duration of dry sampling (< 5 minutes) out of total 60 minutes of sampling.
Second concern associated with the evaporation of water is the variable volume of the PM suspensions collected after different sampling runs depending on the ambient RH, i.e. larger volume at high RH, while smaller volume at low RH. Note, evaporation



can still continue (although substantially slower than during mist formation) after the water level drops below the capillary level. Based on the design of the MC used in this study, final volume of the PM suspension remaining inside the reservoir after the water level drops just below the capillary should be 1.75 mL. However, the volume of PM suspensions obtained from MC during the field sampling was found to vary from 1.5 to 2 mL (i.e. within ±15 % of the theoretically designed volume of 1.75

mL; as observed from numerous trials in extreme RH conditions). We suspected that a variation in the sample volume could potentially affect the DTT activity measurements in RV. To quantify this effect, we conducted a laboratory study, where we measured the DTT activity of a PM extract in different reaction volumes, but all containing the same amount of PM mass. Six circular punches (4.9 cm$^2$ each) from a $PM_{2.5}$-laden HiVol quartz filter (24-hour sample collected from the same site) were extracted in 35 mL of Milli-Q water by sonication for 30 minutes. The extract was filtered by passing through a 0.45 μm PTFE

syringe filter (Fisherbrand™, Fisher Scientific, PA, USA). 0.875 mL aliquot of this filtered extract was added in 12 different sample vials, which ensured the same $PM_{2.5}$ mass in each vial. To the reference sample vial, 0.875 mL Milli-Q water was added to make the total sample volume as 1.75 mL (equivalent to the theoretically designed volume of the PM suspension obtained from MC after one hour of sampling). To the remaining sample vials, different amount of Milli-Q water (0, 0.175, 0.35, 0.525, 0.7, 1.05, 1.225, 1.4, 1.575 and 1.75 mL) were added to obtain variable sample volumes (0.875, 1.05, 1.225, 1.4,

1.575, 1.925, 2.1, 2.275, 2.45 and 2.625 mL). This experiment simulated the changes in the volume of PM suspensions obtained from MC due to evaporation. The DTT activities of all these samples were then measured (in triplicates) using the semi-automated system in our lab. The same experiment was repeated for three more $PM_{2.5}$ filters and all the results are compiled in Fig.S3 (SI). As expected, there is an overall inverse trend in the DTT activity versus sample volume; samples containing larger volumes showed lower DTT activity than those containing smaller volumes. However, maximum bias in the activity for

a 20% variation in the sample volume was less than 6 % (average 3 ± 3 %); therefore, this small bias was neglected for the purpose of this study. Note, a larger variation in the sample volume could significantly affect the measurement of DTT activity (e.g. bias > 12 % for a 50 % variation in the sample volume, as shown in Fig.S3). Therefore, an appropriate control, e.g. a nafion or a diffusion dryer should be used upstream of MC at locations where the ambient RH can vary substantially and more abruptly, to minimize its impact on the variations in sample volume and subsequent bias in the DTT activity measurement.

**3.3 Blanks and limit of detection (LOD)**

The average (±1σ) system and method blanks obtained by the instrument were 0.32 ± 0.06 and 0.33 ± 0.08 nmol/min, respectively, and the difference between these two blanks was not statistically significant (p > 0.05; unpaired $t$-test). Therefore, LOD of the instrument for measuring the PM-induced DTT activity was determined as 0.24 nmol/min, i.e. 3 times the standard deviation of the method blanks. The insignificant difference between the system and method blanks indicates that the ambient

gases (e.g. ozone) have minimal interference in the DTT activity measured by the online instrument. Ozone has very low solubility in water (Henry's law constant ~ 1.03 *$10^{-2}$ M/atm ) (Sander, 2015), which makes it unlikely to get collected in MC. To further confirm the insignificant contribution from ambient gases in the DTT activity, we connected a custom-built activated charcoal denuder [22" long and 3" diameter tube (with a concentric meshed tube for airflow), filled with 4 mm pellet activated



charcoal, Enviro Supply & Service, Irvine, CA], upstream of MC. The field blanks collected for a day (N = 24) with an upstream denuder (i.e. HEPA filter + denuder) were not significantly different ($0.33 \pm 0.09$ nmol/min) than without the denuder (i.e. only HEPA filter). A possible explanation for the nil contribution from ambient gases in our DTT activity measurement could be their low ambient concentrations and a very low residence time (72 ms) in our MC, leading to their insignificant

concentration in the MC suspension.

## 3.4 Online instrument results versus filters measurements

Figure 2 shows a comparison of the results obtained by the online instrument with the DTT activity measured on the water and methanol extracts of the filters (47 mm), which were simultaneously collected while running the instrument. The online measurements from the instrument have a high coefficient of determination ($R^2 > 0.80$) with the results obtained by offline

filter extraction method, both using Milli-Q water and methanol. The slope for the water extraction versus online measurement was only 0.52 ($R^2 = 0.87$; $p < 0.001$; N = 20), while methanol extracts of the PM filters showed a much better agreement with the online system (slope = 0.86; $R^2 = 0.93$; $p < 0.001$; N = 16). This is probably due to significant contribution from some water-insoluble PM components in MC, which otherwise remain embedded in the filter fibers and are poorly extracted in water by sonication. Recently, Gao et al. (2017) also reported a ratio of the DTT activity measured on the water-soluble extracts of

the ambient PM filters collected from Atlanta to the total OP (i.e. DTT oxidation performed directly on the filter) as 0.58 - 0.65. Additionally, the ratio of the DTT activity measured on methanol-soluble extracts of the filters to the total OP was 0.9 - 0.94. The higher OP measured directly on the filters was attributed to the contribution from water-insoluble PM fraction remained on the filters, which is not fully extracted even by methanol. Collectively, our results, which are very similar to Gao et al., (2017), demonstrate a much better efficiency of the MC for PM collection than a conventional filter sampling and

extraction protocol.

## 3.5 Time-series of the ambient PM$_{2.5}$ DTT activity

Figure 3 shows time-series of the hourly ambient PM$_{2.5}$ DTT activity (blank-corrected) measured by the online instrument

between May 31, 2017 and August 16, 2017. The instrument was almost continuously operated for collecting the ambient data (except few days for method or denuder blanks and calibration of the analytical part) between May 31 and July 11, 2017, followed by a break of 22 days (July 12, 2017 - August 3, 2017), before running it again for 13 days in the month of August, 2017. The average ($\pm 1$ $\sigma$) DTT activity obtained during the entire sampling period was $0.33 \pm 0.19$ nmol/min/m$^3$, which is in typical range of the total OP as observed in Atlanta ($0.2 - 0.4$ nmol of DTT/min/m$^3$; Gao et al., 2017) and is higher than the

water-soluble OP measured at the same site ($0.04 - 0.18$ nmol of DTT/min/m$^3$) in our previous studies (Xiong et al., 2017; Wang et al., 2018; Yu et al., 2018). Around 65% of the hourly measurements of DTT activity were above the instrument LOD. However, measurements during the weekends were mostly below the LOD. The low activity over the weekends is possibly



due to less vehicular traffic resulting in lower $PM_{2.5}$ emissions. Additionally, very low DTT activity (or below LOD) was measured on days with exceptionally high RH (> 90 %) and during rain events, probably due to reduced $PM_{2.5}$ concentrations (Lou et al., 2017). A significant diurnal variation in the DTT activity can also be inferred from the time-series plot, with nighttime measurements generally lower than the daytime levels (except few peaks, e.g. on 7/4/2017 and 7/5/2017). A more

detailed analysis of the diurnal trend of DTT activity is provided in section 3.6.

The fireworks display during the Independence Day (July 04, 2017) celebration in Champaign County had a pronounced effect on the ambient $PM_{2.5}$ OP. The DTT activity measured on the night of July 04 and following 2 days, showed a significantly elevated value (peak value = 1.4 nmol/min/m³), which was approximately four times the average DTT activity obtained during the entire sampling period. This is probably due to an overall increase in the ambient $PM_{2.5}$ concentration emitted from fire-

crackers. The $PM_{2.5}$ concentration measured at a nearby Environmental Protection Agency (EPA) monitoring site (Bondville, located ~ 8 miles from the study area) reported an increase of 124 %, 42 % and 49 % in the average $PM_{2.5}$ concentration between July 04 and July 06 (average of three days = 14.0 ± 1.2 µg/m³), compared to July 2 (6.27 µg/m³), July 3 (9.9 µg/m³) and July 7 (9.4 µg/m³; assumed control days), respectively. Pervez et al. (2016) reported a 13- and 7-fold increase in ambient Cu and Mn concentration, in the local background $PM_{2.5}$ after a firework pollution episode in Bhilai, India. Note, Cu and Mn

are among the most important metals in PM, which have been reported to be associated with the DTT activity (Charrier and Anastasio, 2012).

### 3.6 Diurnal profile of the ambient $PM_{2.5}$ OP and chemical components

One of the primary motivations for developing the online instrument was to discern the diurnal variations in ambient $PM_{2.5}$

OP, so that it could be better linked with the chemical components and their emission sources. Therefore, hourly data obtained by running the instrument for 28 days (between May 31 and July 2, 2017) was composited to obtain a diurnal profile of the DTT activity as shown in Fig.4. The diurnal profiles were separately plotted for the weekdays and weekends in Fig.4a and 4b, respectively. The diurnal profile of OP in weekdays appears to have a trimodal trend with peaks at 7:00 – 8:00 AM, 12:00 – 1:00 PM, and 6:00 – 7:00 PM. The highest peak was observed around 1:00 PM, and night time nadir was around 3:00 AM. In

contrast, the weekend diurnal profile appears to be flat with no clear trend. To test if the identified peaks in the diurnal profile of DTT activity are statistically significant (given the large variation in day-to-day activity), we first performed a single factor ANOVA test [SPSS (V24.0.0.0)] on all points of the diurnal plot (N = 24). The diurnal profile for the weekends did not yield any statistically significant peak (F=1.51, F $_{critical}$ = 1.59, p = 0.07) for the entire data. However, F value obtained for the weekday diurnal profile data was higher than the critical F value (F = 6.38, F $_{critical}$ = 1.56, p < 0.01) indicating significant

differences in certain hourly measurements during weekdays. To further test the statistical significance of the three identified peaks in the diurnal profile for weekdays, we conducted a Student's *t*-test between different pairs of the hourly data. For example, all 7:00AM measurements were compared with 5:00 AM and 9:00 AM measurements; 1:00 PM measurements were compared with 9:00 AM and 4:00 PM measurements, and 7:00 PM peak was compared with 1:00 PM, 4:00 PM and 9:00 PM data, individually. The measurements at 7:00 AM were found to be statistically different from 5:00 AM and 9:00 AM (p <



0.05). Similarly, the measurements at 9:00 AM and 1:00 PM were also statistically different ($p < 0.01$). Although, the difference among the 1:00 PM, 4:00 PM and 7:00 PM data was not statistically different ($p = 0.16$ for t-test between 1:00 PM and 4:00 PM, 0.45 between 4:00 PM and 7:00 PM and 0.58 between 1:00 PM and 7:00 PM), the 7:00 PM and 9:00 PM measurements were statistically different ($p < 0.05$). Thus, it was inferred based on these tests that the diurnal OP trend is actually bimodal

with the first peak during the morning period, followed by an afternoon peak, which flattens out (i.e. extends from 1:00 PM to 7:00 PM, but with a slight dip at 4:00 PM), before subsiding at night (around 9:00 PM).

To further investigate the diurnal profile of the DTT activity and its association with various emission sources at the site, we analyzed chemical composition of $PM_{2.5}$ collected on eight time-segregated composite filters, i.e. QS1-QS4, and TS1-TS4 in August. The schedule of this time-segregated sampling was determined based on the diurnal profile of the DTT activity in

weekdays (Fig.4). Note, the diurnal profile in DTT activity obtained by the instrument during the time-segregated sampling (Fig.S4 in SI), had a similar trend as obtained in the month of June (Fig.4). We measured the concentration of key chemical species, i.e. WSOC, OC, EC and transition metals (Cu, Mn and Fe), which all have been shown to be associated with the DTT activity (Charrier and Anastasio, 2012; Saffari et al., 2014; Verma et al., 2014; Yang et al., 2014) in our time-segregated samples, along with the PM mass, to infer their diurnal variations and compare it with that in the DTT activity. The average

$PM_{2.5}$ concentration during the time-segregated sampling was $7.8 \pm 0.16 \, \mu g/m^3$, $8.5 \pm 0.36 \, \mu g/m^3$, $8.1 \pm 0.16 \, \mu g/m^3$ and $7.6 \pm 0.29 \, \mu g/m^3$ (not shown) for the morning, afternoon, evening, and night samples, respectively. Given, there are no substantial variations in $PM_{2.5}$ mass concentrations, the diurnal variations in the DTT activity are hypothesized to be mainly attributed to the varying chemical composition of $PM_{2.5}$ resulting from different emission sources at the site.

**3.6.1 Carbonaceous aerosol**

Figure 5 shows $PM_{2.5}$ carbonaceous content, in terms of the ambient concentrations of EC, OC and WSOC during time-segregated sampling. EC can be assumed as a marker of vehicular emissions (Shah et al., 2004; Shirmohammadi et al., 2016). EC showed a peak in the morning and also at night. The morning peak coincided with the rush hour traffic and can be attributed to the vehicular emissions. It then subsides in the afternoon and remains constant till evening, before getting elevated again at

night. Based on our personal observation, vehicular traffic is almost negligible after 10:00 PM at this location. Therefore, EC peak observed at night is probably a result of the decrease in atmospheric mixing height at the site (Liu and Liang, 2010). The lower temperature and higher relative humidity [average values = 19°C and 78 % at night (7:00 PM-7:00 AM) compared to 27°C and 48 % in daytime during the time-segregated sampling] might have resulted into limited atmospheric mixing at night. Furthermore, EC has been reported to be distributed uniformly in the vertical direction and is thus greatly influenced by the

atmospheric mixing height (Lin et al., 2009; Mues et al., 2017). A similar pattern in EC (i.e. peaks in morning and late evening) has been reported in few other studies (Mues et al., 2017; Sharma et al., 2017; Singh et al., 2018), where the lower concentrations in daytime were attributed to a higher mixing height and stronger turbulence than at nighttime. Note, our nighttime composite filter was sampled from 7:00 PM – 7:00 AM, thus probably including part of the EC from evening rush hour traffic, which is trapped in the lowered mixing height at night. Overall, the diurnal trend in EC does not follow the trend





in DTT activity, which indicates a negligible role of EC in determining the DTT activity at this location. These results contrast with earlier studies showing a high correlation between EC and DTT activity (Verma et al., 2012; Gao et al., 2017). However, it should be noted that EC concentrations measured in this study ($0.15 - 0.24$ μg/m$^3$) were much lower than the values reported by other studies conducted at similar urban locations. For instance, Verma et al. (2014) reported monthly averaged summer

EC concentration of $0.75 \pm 0.25$ μg/m$^3$ in Atlanta, while the concentrations in the range of 0.4 - 0.52 μg/m$^3$ were reported for the samples collected in Los Angeles during summer 2012 (Shirmohammadi et al., 2016). Therefore, EC concentration at our sampling location is probably too low to make a significant contribution in the DTT activity. Additionally, fresh EC is less active and needs to be oxidized to participate in the DTT assay, as indicated by Antiñolo et al. (2015), which reported an almost exponential increase in the DTT activity of soot particles after exposure to high levels ($>10^{12}$ molecule cm$^{-3}$ h) of ozone.

Given the PM$_{2.5}$ collected at the site, which is adjacent to a roadway, contain mostly fresh and unoxidized EC, it probably leads to a poor association between the diurnal trends in EC and DTT activity in our study.

    The secondary organic aerosols (SOA) formed by photochemical reactions has been reported as one of the major contributor to the ambient PM$_{2.5}$ OP (Snyder et al., 2009; Verma et al., 2009, 2014; Saffari et al., 2015). WSOC is a substantial component of SOA formed during photochemical reactions (Cheung et al., 2012; Verma et al., 2014). On the other hand, OC is emitted

from both primary (i.e. combustion) and secondary (e.g. photochemical formation) sources (Cabada et al., 2004). As depicted in Fig.5, neither OC nor WSOC show any diurnal pattern. Figure 5 also shows OC/EC ratio, which peaked in the afternoon. Considering a higher mixing height and relatively lower traffic in the afternoon than morning, an elevated OC/EC ratio indicates an additional OC contribution, which compensates its decrease from reduced vehicular emissions and enhanced atmospheric mixing. We attribute this additional OC to the secondary particle formation via photochemical reactions, which

keeps the OC concentration almost constant throughout the day. Figure S5 in SI shows a diurnal profile of ozone measured at Bondville (EPA site). The ozone concentration peaked from 11:00 AM to 6:00 PM indicating secondary formation of particles in the afternoon period. To further confirm the contribution from SOA to OC, WSOC/EC ratio was also plotted (Fig.5), which ranged from 6.6 (morning) to 11.3 (evening) and followed a similar diurnal profile as ozone or OC/EC ratio. Thus, the broad peak in DTT activity during afternoon and evening periods could partly be caused by the redox-active SOA components.

**3.6.2 Trace metals**

    Vehicular sources, which include both direct exhausts and non-tailpipe emissions (e.g. brake and tire wear, abrasion of the road surface and resuspension of road dust) contribute majorly to the concentration of trace metals in the atmosphere (Thorpe and Harrison, 2008). Hulskotte et al. (2007) identified brake wear as one of the main sources of airborne Cu emission. In addition, resuspended road-dust is a mixture of various elements including the redox-active transition metals (e.g. Cu, Mn and

Fe) (Wang et al., 2005; Thorpe and Harrison, 2008; Chen et al., 2012). Figure 6 shows the ambient concentrations of three metals, i.e. Cu, Mn and Fe measured at the sampling site. The concentrations of Cu varied between 4 and 22 ng/m$^3$, and was comparable to the concentrations reported in other studies conducted in urban areas (Verma et al., 2009, 2014). However, the concentration of Fe (3.68 - 6.85 ng/m$^3$) was significantly lower than reported in other studies. Fe concentrations typically found



in urban atmosphere are in the range of 90 – 280 ng/m$^3$ (Verma et al., 2009, 2014; Xia and Gao, 2011). The concentration of Mn at this site was very low ($\leq 1.1$ ng/m$^3$) during the entire sampling period.

Both Cu and Fe followed a very similar diurnal trend as the DTT activity (Fig.4), with highest concentration in the afternoon and lowest at night. The difference in the diurnal trends of metals and EC concentrations (i.e. EC peaks in the morning while

metals peak in the afternoon) indicates a lesser influence of direct vehicular exhausts on the metals concentrations than the resuspended dust, which is generally driven by higher vehicular speeds (due to relatively lower traffic) and drier conditions in the afternoon (Pant and Harrison, 2013). Figure S6 in SI shows the diurnal pattern of ambient RH at the sampling site. The very high RH ($> 75$ %) substantially suppresses the dust resuspension and the resultant metals concentrations during nighttime. Among all the metals, Cu is known to be the strongest oxidizer of DTT (Charrier and Anastasio, 2012). Therefore, a close

similarity of the diurnal profile of Cu with DTT activity indicates a significant contribution of both directly emitted Cu from vehicular exhausts (the morning peak) and resuspended Cu (in the afternoon) to PM$_{2.5}$ OP at the sampling site.

## 4 Conclusions

We developed and field-tested a low-cost online instrument for the near real-time measurement of ambient PM$_{2.5}$ OP with an hourly resolution. The instrument is based on coupling an automated analytical system for DTT activity determination to an

MC designed to collect PM$_{2.5}$ suspension in water. The MC, equipped with a hydrophobic filter collects PM$_{2.5}$ suspension by generating a fine mist of water, which scrubs the particles, both directly from air and those collected onto the filter. The PM suspension collected in MC is then fed to an automated analytical system for the measurement of DTT activity.

The OP data collected using the online instrument showed strong correlation (R$^2 > 0.8$; $p < 0.001$) with the measurements obtained by the conventional filter sampling and extraction (in water and methanol) methodology. However, the online

instrument yielded higher DTT activity than both water (slope = 0.52, N = 20) and methanol (slope = 0.86, N = 16) extracted filters. We attribute this higher DTT activity to the contribution from water-insoluble components, which are more effectively collected in MC than the conventional filter sampling and extraction protocol. The instrument was tested in the field for 50 days in summer 2017 with minimal manual assistance. The instrument's LOD as tested under field condition is 0.24 nmol/min and around 65% of the hourly DTT activity measurements during field sampling were above the LOD. Substantially lower

DTT activity levels (i.e. either below or very close to LOD) were observed on the days with very high RH ($> 90$%) or rain events. Interestingly, the DTT activity showed a four-fold increase on the night of July 04 (Independence Day), caused by the fireworks display, as compared to the average OP measured during the entire sampling period.

The hourly data collected during field sampling allowed us to plot the first-ever recorded diurnal profile of the DTT activity of ambient PM$_{2.5}$. The diurnal profile for weekdays showed a bimodal trend with first peak in the morning (around 7 AM),

followed by a broad peak (extending from 1 PM to 7 PM, but with a slight dip at 4 PM) during afternoon and early evening periods. We further investigated this bimodal profile of ambient PM$_{2.5}$ DTT activity in relation to the diurnal variation in PM$_{2.5}$ chemical composition, by collecting time-segregated composite filters for 10 consecutive weekdays. By comparison of DTT activity with various chemical components, i.e. OC, EC, WSOC, Cu, Fe and Mn, the morning peak in DTT activity profile



was attributed to the metals emissions from vehicular sources (exhausts and non-exhausts), whereas both secondary formation (i.e. SOA) and metals from resuspended dust seem to contribute to the afternoon peak. Overall, the extensive field-testing of the instrument in our study demonstrates its stability and consistency to yield the long-term measurements of ambient PM$_{2.5}$ OP. Further studies in this direction should be aimed to integrate the highly time-resolved data obtained by online OP and other real-time instruments for measuring the PM$_{2.5}$ chemical composition, and thus infer the diurnal contribution of various emission sources in the risk associated with PM$_{2.5}$ exposure.

*Competing interests*: The authors declare that they have no conflict of interest.

*Acknowledgments*: This work was supported from V.V.' s startup fund by the Department of Civil and Environmental Engineering at University of Illinois Urbana− Champaign. We thank Michelle Wahl (parking director) for providing us the site for measurement.

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





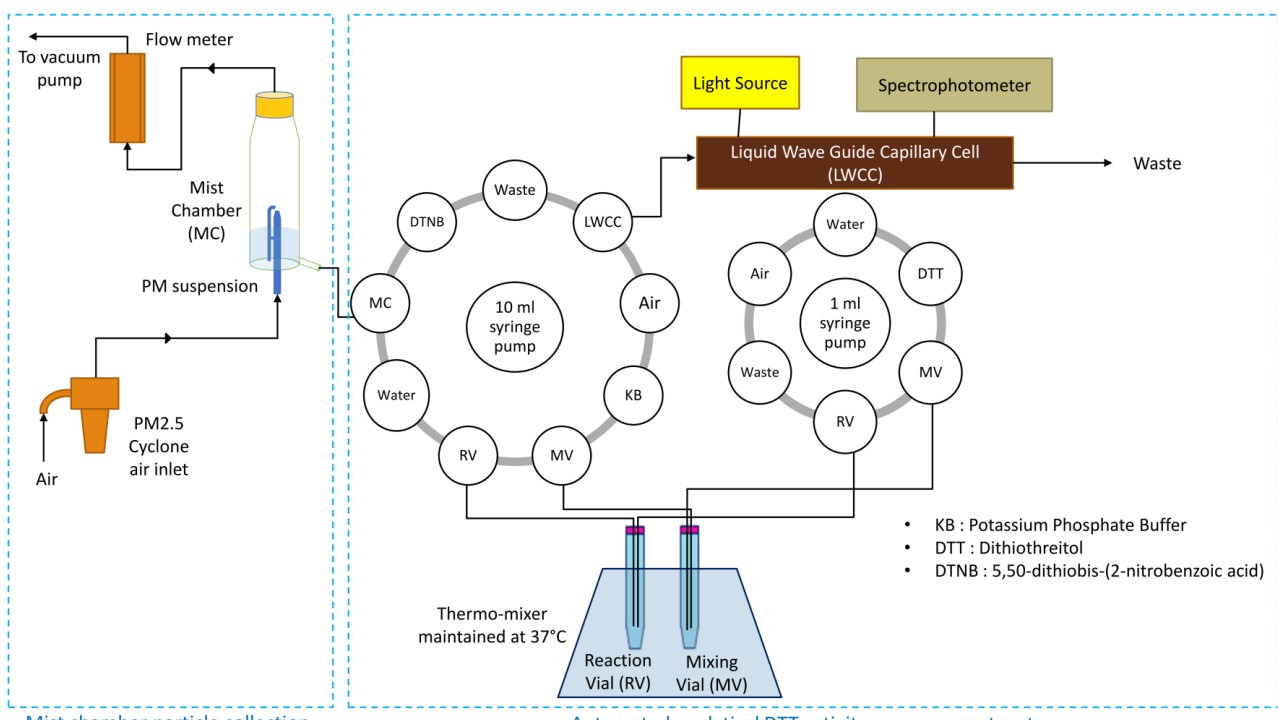

Figure 1. Layout of the online instrument for measuring the DTT activity of ambient $PM_{2.5}$





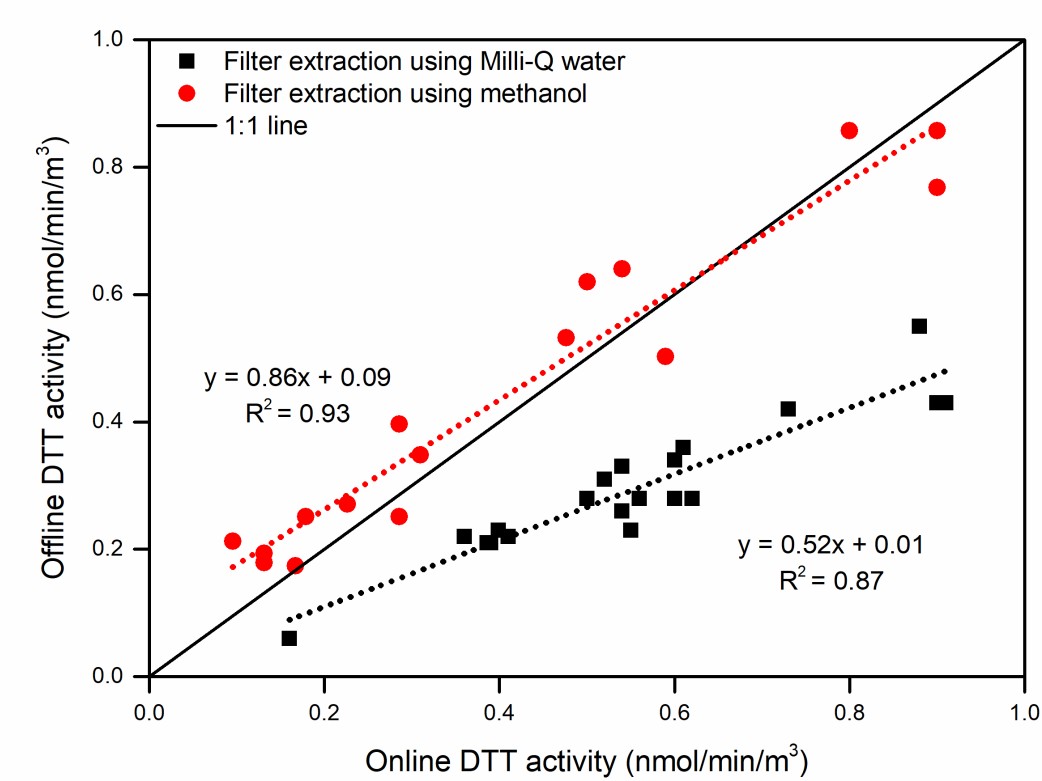

Figure 2. Comparison of the DTT activity obtained from the online instrument with the traditional

5    filter collection and extraction methodology (i.e. offline DTT activity analysis of the PM extracts)





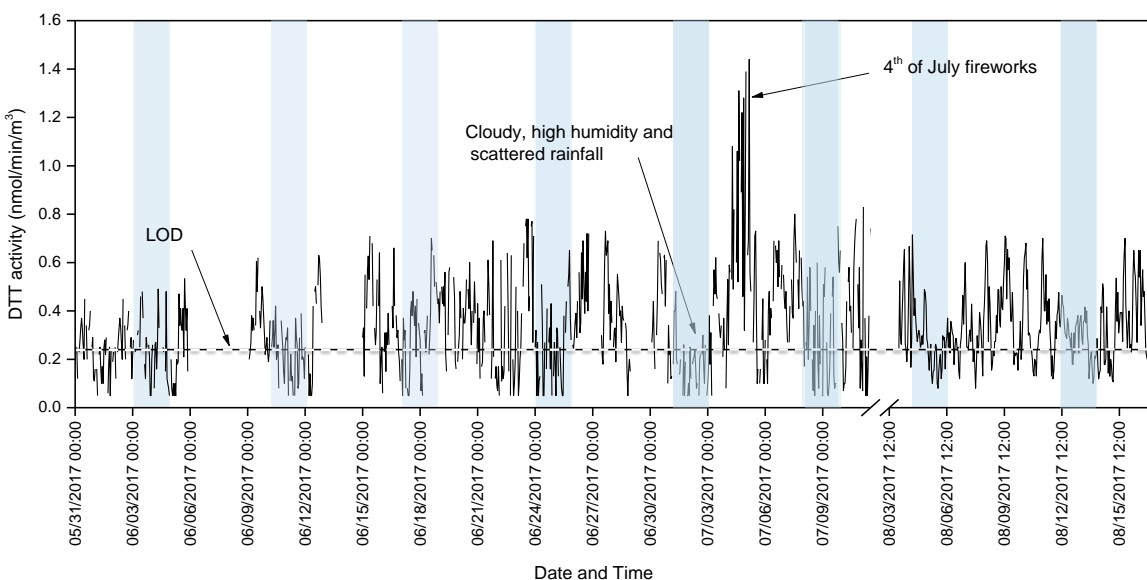

Figure 3. Time-series plot of the DTT activity from May 31,2017 to Aug 16, 2017. The shaded portions in the graph are weekend DTT activity measurements. The instrument was not operated between May 31 and

5                                                                July 11, 2017 (shown as a break in the X-axis).





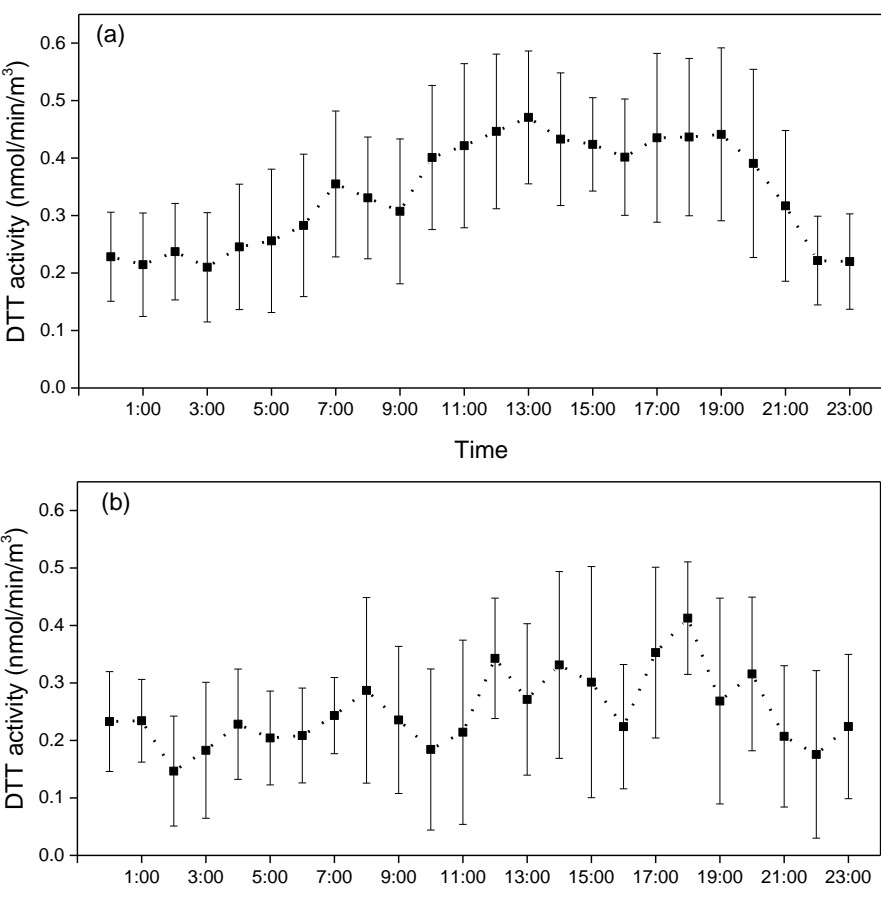

5    Figure 4. Diurnal profile of the ambient PM$_{2.5}$ DTT activity measured at the sampling site over (a)
weekdays (n = 18 days), (b) weekends (n = 10 days). The data from May 31 to July 2, 2017 was
used for plotting this profile. Error bars are the standard deviation (1σ) of the average DTT activity
in that hour.



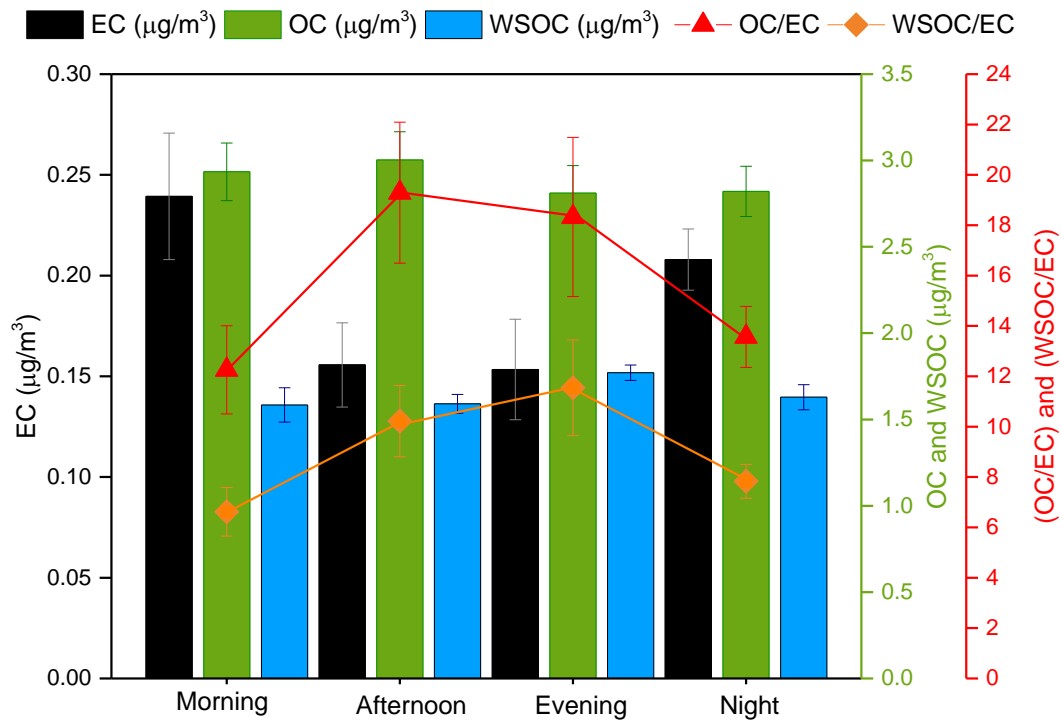

Figure 5. Diurnal trend in the ambient concentrations of carbonaceous aerosols, i.e. EC, OC, and
WSOC, and the ratio of OC/EC and WSOC/EC, as measured from the time-segregated integrated
filters. Error bars denote standard deviation (1 σ) of the triplicate measurements.




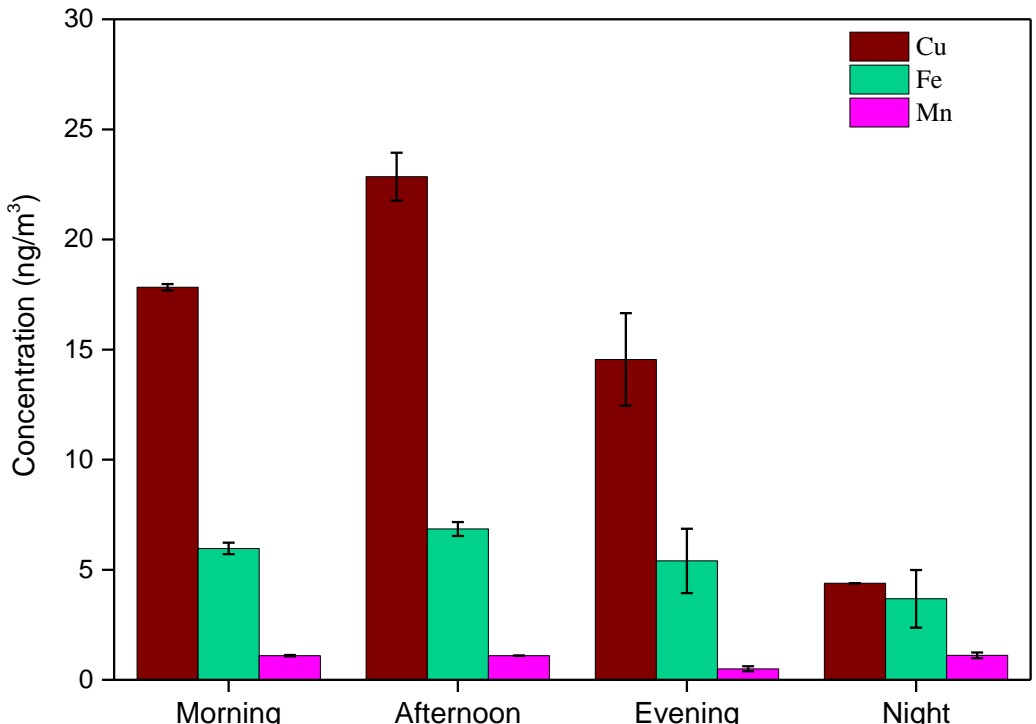

Figure 6. Diurnal trend in the ambient concentrations of metals, i.e. Cu, Fe and Mn, as measured from the time-segregated integrated filters. Error bars denote standard deviation (1 σ) of the triplicate measurements.