# Peer review of "Development and field-testing of an online instrument for measuring the real-time oxidative potential of ambient particulate matter based on dithiothreitol assay"

_Atmospheric Measurement Techniques, 2018_

## Referee Comment (RC1) · Anonymous Referee #3 · 30 Jul 2018

Review of "Development and field-testing of an online instrument for measuring the real-time oxidative potential of ambient particulate matter based on dithiothreitol assay" by J. V. Puthussery et al.

This study describes the development of an automated, on-line system to measure the oxidative potential (OP) of particulate matter using the DTT assay. Measurements of OP and DDT activity have been increasing substantially in recent years with the acknowledgement that this may represent an indicator of the toxicity of PM. The system described in this manuscript represents the first automated measurement of DTT activity, and is therefore an important contribution to the atmospheric measurement community. Overall, the authors present a detailed, careful, and deliberate characterization of the system performance. The extensive ambient deployment of the system goes beyond most method papers. The writing is clear, the paper is well organized, and the figures are of high quality. I have several issues with the manuscript: I see some problems with the authors' attribution of the DTT activity to specific sources – it seems they are trying to explain their results based on what they expect, which does not necessarily agree with their actual observations. I also think that the different contributions of insoluble particles to the mist chamber samples is a bigger deal than the authors describe, and certainly contributes to the differences in OP between the offline and online measurements, but also to their troubles attributing the DTT activity to specific sources. I recommend the manuscript for publication after my comments have been addressed.

**Specific Comments:**

I assume that this has been addressed in the references cited, but some discussion of the particle collection efficiency by the mist chamber is needed.

The difference between the MC ROS and filter ROS is quite surprising (50% higher in the on-line system), and definitely indicates the contribution of insoluble components. Some discussion of Phillips and Smith (2017) should be included here. I think the major implication of this is that a direct comparison of the MC and filter samples is not so straightforward. The MC samples include a lot of insoluble particles which clearly contribute to the DTT activity (similar to the MeOH extracted samples), which is not the case for the water soluble extracts. This muddles the comparison of the DTT activity to the other chemical components.

Pg. 9, line 18 – 20: I disagree with this statement. The collection of insoluble particles does not indicate that the MC performs better than conventional filter collection and extraction, but rather that the MC is subject to an artifact that may need to be accounted for.

Discussion on Pg. 2, lines 25 – 27: I agree there is great utility in an on-line measurement of oxidative potential, but this argument is rather weak. If Cu is really the most important species driving DTT activity, is there any evidence that Cu undergoes chemical changes during filter sampling?

Pg. 3, line 5: I do not understand this sentence?

Pg. 3, lines 17 – 20: this is somewhat misleading, since the ancillary components (syringe pumps, distribution valves, LWCC, spectrometer) will in total cost many thousands of dollars. My guess at the total system cost is $12k - $16k, and while this represents a lower cost than a method using the PILS or LSS, I don't think this qualifies as "low-cost".

Section 2.2: give the manufacturer and purity of all chemicals and reagents.

Section 3.2: the strong RH-dependence on the remaining water volume is not consistent with other M.C. studies (e.g., Hennigan et al., 2018) – this may be due to inconsistencies between different mist chambers, but this difference is worth mentioning.

I really do not understand the results in Fig. S3 (and associated discussion in Pg. 8 lines 10 – 25). Why does a 50% change in the sample volume (either + or -) not result in a 50% change in the DTT activity measurement?

Pg. 12, line 10 – 11: I don't agree with this logic – the authors state that traffic at their site after 10pm is almost nonexistent, which would suggest some of the EC is aged.

Pg. 12, line 23 – 24: this does not seem consistent with the data in Figure 5. The WSOC diurnal profile is flat throughout the day. Most (if not all) of the WSOC during summer at this location should be secondary – how then do the authors attribute the afternoon increase in DTT to SOA?

Pg. 13, lines 4 – 11: I disagree with this interpretation of the data. I do not think that Figures 4 and 6 show a strong contribution of Cu to the measured OP. See prior comments, but it's important to acknowledge that Figures 4 and 6 cannot be directly compared due to the contribution of insoluble particles to the Fig. 4 measurements.

**Technical Corrections:**

Pg. 1, line 9: "using the dithiothreitol…"

Pg. 1, line 17: date format should be 4 July

Pg. 2, line 17: delete comma before 'PM'

Pg. 4, line 28: try to avoid beginning a sentence with a number

Figure 3: I believe the incorrect dates are given in the caption for the time the instrument was not operated.

**References:**

Hennigan, et al. Technical note: Detailed characterization of a mist chamber for the collection of water-soluble organic gases, Atmospheric Environment, 188, 12 – 17 (2018).

Phillips, S. M., Smith, G. D., Spectroscopic comparison of water- and methanol-soluble brown carbon particulate matter, Aerosol Science & Technology, 51, 1113 – 1121 (2017).

---

## Referee Comment (RC2) · Anonymous Referee #1 · 8 Aug 2018

In this work, the authors designed a novel instrument to measure oxidative potential in ambient PM2.5 on an hourly timescale. They use a mist chamber to collect particles and direct subject these particles to reaction with dithiothreitol, a model antioxidant to infer the ability of inhaled particles to cause oxidative stress. It represents an important improvement to previous filter-based methods, and while the new measurements show good agreement with measurements made using these methods, they were substantially higher. The instrument is well characterized and carefully designed. I have only one major concern about this manuscript, but I recommend publication in AMT after

addressing this and other minor concerns.

Major comment:

My only major concern is the section on comparison to composition measurements. The comparison was performed on two different time periods. While the authors claim that the composition should be similar between these two time periods, there is also no reason to believe they would be similar. For example, the diurnal variation in DTT activity may be heavily driven by a few anomalous days during July 4/5. This section is considerably weaker than the characterization tests described in the other sections. I also do not see why this comparison needs to be made in this manuscript. I believe it is sufficient to demonstrate the time-resolved capability of this instrument for this manuscript. I suggest removing this section, or re-writing this section, and performing the comparison experiments much more carefully in the future.

Other comments:

Abstract: It is confusing to read that the measurements made by MC are higher than filter-based methods, and then see slopes of less than 1. I suggest reporting the slope of MC vs filter-based method.

Introduction: The authors claim that the MC is substantially cheaper (of $\sim$150 USD) is a vast overstatement. The cost of machining the MC alone would be greater than 150 USD. Also, the authors should consider that university facilities are highly subsidized and the cost that researchers see may be substantially lower than the true cost.

Section 3.2: The tests described in this section are very appropriate. I believe that researchers in this field have largely neglected the importance of reactant concentration in determining reaction rates. Assuming a simple bimolecular reaction between DTT and PM, there should be a linear relationship between reactant concentration and measured DTT consumption, even at the same PM mass. What is puzzling me is that the authors only observe a 6% change for a 20% change in volume (or concentration).

Do the authors have any explanation for this small change? Does this suggest that the DTT reaction with PM is not bimolecular?

Section 3.4: Is there any evidence of water insoluble components in MC samples? For example, can the MC samples be filtered and the DTT measurements be repeated? Which is more biologically relevant? I can imagine the insoluble components remain in the lung lining fluid and continue to consume antioxidants.

Section 3.6.1: The authors argue that EC, Cu and Fe are all coming from traffic sources. However, the diurnal patterns are not consistent with each other. EC is higher in the morning and at night, but Cu and Fe decreases substantially at night. The authors appear to be contradicting themselves here.

Throughout the manuscript, the authors have used OP to describe DTT consumption rate per volume of air, whereas it is also often used to describe DTT consumption rate per mass of PM. I suggest defining it outright in the introduction. The per volume DTT consumption is often referred to as the "extrinsic" oxidative potential, or the oxidative capacity.

Minor technical comments:

Page 2 line 9: times

Page 2 line 10: what does it mean by "less controlled environment". Cell-free assays would be conducted under a more well-controlled environment than cellular assays.

Page 2 line 34: the word "data" is plural; remove "a"

Page 5 line 14: "the" analytical part

Page 5 line 30: "the" analytical part

Page 7 line 19: replace "reduces" with "is reduced" or "decreases"

Page 7 line 31: "The second concern"

Section 3.3: It may be more useful to report LOD in terms of oxidative capacity in nmol DTT/(min-m3 air)

Page 9 line 24: "the time series"

Section 3.6.1: EC might be more a marker of diesel traffic

Page 11 line 24: awkward language: "getting elevated"

Page 12 line 33: there may be too many significant digits for Fe concentrations

Page 24 line 6: again, the word "data" is plural; replace "was" with "were"

---

## Author Comment (AC1) · 31 Aug 2018

**Response to Anonymous Referee #1**

In this work, the authors designed a novel instrument to measure oxidative potential in ambient PM2.5 on an hourly timescale. They use a mist chamber to collect particles and direct subject these particles to reaction with dithiothreitol, a model antioxidant to infer the ability of inhaled particles to cause oxidative stress. It represents an important improvement to previous filter-based methods, and while the new measurements show good agreement with measurements made using these methods, they were substantially higher. The instrument is well characterized and carefully designed. I have only one major concern about this manuscript, but I recommend publication in AMT after addressing this and other minor concerns.

We thank the reviewer for his/her recommendation on the acceptance. Our responses and corresponding changes made in the manuscript (highlighted in red) are given below.

Major comment:

1.My only major concern is the section on comparison to composition measurements. The comparison was performed on two different time periods. While the authors claim that the composition should be similar between these two time periods, there is also no reason to believe they would be similar. For example, the diurnal variation in DTT activity may be heavily driven by a few anomalous days during July 4/5. This section is considerably weaker than the characterization tests described in the other sections. I also do not see why this comparison needs to be made in this manuscript. I believe it is sufficient to demonstrate the time-resolved capability of this instrument for this manuscript. I suggest removing this section, or re-writing this section, and performing the comparison experiments much more carefully in the future.

There seems to be a misunderstanding on the comparison study conducted in this paper which we hope to clear through our response below.

The PM chemical composition data was collected for a period of 10 "weekdays" between August 3 – August 16, 2017. The diurnal profile of the ambient $PM_{2.5}$ extrinsic DTT activity ($OP_{ex}$) obtained during this period is also provided in Figure S4 (supplementary information). Since, the diurnal profile of DTT activity shown in Figure S4 was similar to the diurnal profile obtained during June 2017 (i.e. Figure 4a in the manuscript) (Page 23, line 1), we did not feel it necessary to include both the figures in the main manuscript. Now, given we performed only a *qualitative comparison* (e.g. high during the afternoon, low at night) between the DTT activity and chemical components, comparing the chemical composition data with either profile (i.e. either during August 3 - August 16, 2017 or June 2017) will essentially yield the same conclusion as reported in the manuscript.

Please note that we agree with the reviewer that during the Independence Day celebration week (July 4 - 5), the composition of the PM was probably very different, and the diurnal OP trend would have been significantly influenced by these "anomalous days". *Therefore, we did not include the DTT activity data collected during that period for plotting the diurnal profile in Figure 4*. Rather, we have used the data only from May 31 to July 2, 2017. This has been explained in the caption of Figure 4 and the associated text:

Page 23, line 5

"Figure 4. Diurnal profile of the ambient $PM_{2.5}$ DTT activity measured at the sampling site over (a) weekdays (n = 18 days), (b) weekends (n = 10 days). The data from May 31 to July 2, 2017 were used for plotting this profile. Error bars are the standard deviation ($1\sigma$) of the average DTT activity in that hour."

Page 10, lines 19 - 23

"One of the primary motivations for developing the online instrument was to discern the diurnal variations in ambient $PM_{2.5}$ $OP_{ex}$, so that it could be better linked with the chemical components and their emission sources. Therefore, hourly data obtained by running the instrument for 28 days (between May 31 and July 2, 2017) was composited to obtain a diurnal profile of the DTT activity as shown in Fig.4. The diurnal profiles were separately plotted for the weekdays and weekends in Fig.4a and 4b, respectively..."

Although, the comparison was only qualitative, we believe it adds valuable contribution to the manuscript by providing some insights into the possible emission sources, which could impact the DTT activity. Therefore, we would like to keep this section in the manuscript, hoping the reviewer is now clear on it.

Other comments:

2. Abstract: It is confusing to read that the measurements made by MC are higher than filter-based methods, and then see slopes of less than 1. I suggest reporting the slope of MC vs filter-based method.

We have made the changes suggested by the reviewer

Page 1, lines 12 – 15

"The online DTT activity measurements correlated well with the offline measurements but were higher than both methanol (slope = 1.08, $R^2$ = 0.93) and Milli-Q water (slope = 1.86, $R^2$ = 0.86) extracts of the PM filters, indicating a better efficiency of MC for collecting the water-insoluble fraction of PM."

Page 9, lines 10 – 17

"The slope for the water extraction versus online measurement was 1.86 ($R^2$ = 0.86; $p < 0.001$; N = 20), while methanol extracts of the PM filters showed a much better agreement with the online system (slope = 1.08; $R^2$ = 0.93; $p < 0.001$; N = 16). This is probably due to significant contribution from some water-insoluble PM components in MC, which otherwise remain embedded in the filter fibers and are poorly extracted in water by sonication. Recently, Gao et al. (2017) also reported a ratio of the DTT activity measured on the water-soluble extracts of the ambient PM filters collected from Atlanta to the total $OP_{ex}$ (i.e. DTT oxidation performed directly on the filter) as 1.54 – 1.72. Additionally, the ratio of the DTT activity measured on methanol-soluble extracts of the filters to the total $OP_{ex}$ was 1.06 - 1.1."

Page 13, lines 19 – 20

"However, the online instrument yielded higher DTT activity than both water (slope = 1.86, N = 20) and methanol (slope = 1.08, N = 16) extracted filters."

Figure 2 has been modified as:

[Figure]

Figure 2. Comparison of the DTT activity obtained from the online instrument with the traditional filter collection and extraction methodology (i.e. offline DTT activity analysis of the PM extracts)

3. Introduction: The authors claim that the MC is substantially cheaper (of ~150 USD) is a vast overstatement. The cost of machining the MC alone would be greater than 150 USD. Also, the authors should consider that university facilities are highly subsidized and the cost that researchers see may be substantially lower than the true cost.

We agree with the reviewer. Our intention here was to compare the MC system with other existing sample collection systems. Therefore, we have omitted the "low cost" term from the manuscript and removed the cost comparison discussion with other online systems.

Page 3, lines 16 - 21

"Here, we discuss the development of an automated online instrument to measure the hourly averaged OP of ambient $PM_{2.5}$ using DTT assay and its evaluation in the field conditions for over 50 days. A custom-built glass MC was used for collecting the ambient PM suspension, which is then transferred to an automated analytical system for DTT activity determination."

4.Section 3.2: The tests described in this section are very appropriate. I believe that researchers in this field have largely neglected the importance of reactant concentration in determining reaction rates. Assuming a simple bimolecular reaction between DTT and PM, there should be a linear relationship between reactant concentration and measured DTT consumption, even at the same PM mass. What is puzzling me is that the authors only observe a 6% change for a 20% change in volume (or concentration). Do the authors have any explanation for this small change? Does this suggest that the DTT reaction with PM is not bimolecular?

Please see our detailed response to a similar comment raised by another reviewer (anonymous referee 3 comment 9). For the specific comment of the reviewer on bimolecular reaction between DTT and PM, we would like to clarify that the reaction is not between DTT and PM but rather between DTT and oxygen (which is in excess), and PM acts as a catalyst (Cho et al., 2005; Sauvain and Rossi, 2016). The various chemical components in PM, e.g. Cu, Mn, oxygenated organic compounds and quinones have a distinct (and not necessarily linear) response function with the DTT oxidation rate and can be considered as the particle's characteristics (see Charrier and Anastasio, 2012; Sauvain and Rossi, 2016). Recently, Yu et al. (2018) also reported potential synergistic and antagonistic effects of different chemical components in the PM on its DTT response. Thus, the total DTT activity of a PM sample is due to the combined response function from its various components. Furthermore, note that PM extract accounts for only 70% of the total volume in the reaction vial. That means, a 20% volume increase in the PM extract would result only into a 14% increase in the reaction volume, and a 12 % decrease in the concentration of DTT. The initial concentration of DTT also affects its self-oxidation rate as shown by Sauvain and Rossi (2016). Given these complexities, we tried to empirically derive the effect of change in the sample volume on the DTT activity measurements by conducting the experiments using four different PM samples. The difference in the DTT activity measurement bias (shown as error bars in Figure S3) is probably due to differences in the chemical composition of these samples. However, our main objective for performing this experiment was to determine the maximum acceptable variation in the sample volume from MC that leads to a negligible bias in the DTT activity measurement. We found that for a 20% variation in the sample volume, the % bias in DTT activity measurement was less than 6 %; which was neglected for the purpose of this study. We cannot comment on the DTT assay mechanism based on this experiment. A more systematic study with different standard chemicals and a larger sample set of ambient PM samples is probably needed to investigate and provide an exact reaction mechanism of PM in DTT assay, which is beyond the scope of the current study.

5. Section 3.4: Is there any evidence of water insoluble components in MC samples? For example, can the MC samples be filtered and the DTT measurements be repeated? Which is more biologically relevant? I can imagine the insoluble components remain in the lung lining fluid and continue to consume antioxidants.

There is no direct evidence for the collection of water-insoluble components in MC samples. MCs have traditionally been designed to collect the ambient water-soluble PM components and gases. Our study is probably the first to indicate the collection of insoluble PM components in MC. Although, we have not performed any experiments after filtering the MC samples, we observed the DTT activity of these samples to be almost similar to the DTT activity measured after extracting the filters in methanol (Figure 2). Based on this, we have concluded that MC was probably collecting part of the water-insoluble PM fraction, which is contributing to the DTT consumption. We acknowledge the reviewer's suggestion of filtering the MC sample, and we will attempt to include it in our future investigations for a systematic determination of the contribution of insoluble fraction by comparing the OP of filtered and unfiltered MC PM suspensions (i.e. running 2 MCs in parallel), but it is practically infeasible to include this analysis in the current manuscript.

We are not in position to answer the second part of the reviewer's comment on the relative importance of soluble versus insoluble components. Although, it is biologically plausible that water-insoluble components remain in the lung lining fluid for a longer period and continue to consume antioxidants, this hypothesis has not been tested in the toxicological studies, which were traditionally more focused on assessing the water-soluble PM components. There are definitely growing evidences on the biological relevance of water-insoluble PM components with recent studies showing a substantial fraction of the PM OP stemming from the water-insoluble fraction (Akhtar et al., 2010; McWhinney et al., 2011; Verma et al., 2012; Li et al., 2013; Gao et al., 2017). Therefore, it is important to account for this fraction when determining the overall OP of inhalable PM and incorporating it in the toxicological and epidemiological studies assessing the health effects of PM.

6. Section 3.6.1: The authors argue that EC, Cu and Fe are all coming from traffic sources. However, the diurnal patterns are not consistent with each other. EC is higher in the morning and at night, but Cu and Fe decreases substantially at night. The authors appear to be contradicting themselves here.

We disagree with the reviewer about the contradiction in our interpretation. EC is directly emitted from the vehicular exhausts (mainly from diesel vehicles) (Shah et al., 2004), while water soluble Cu and Fe are emitted from both exhausts and non-exhaust components of the vehicular emissions, i.e. brake and tire wear, and abrasion of the road surface (Thorpe and Harrison, 2008). In addition to that, resuspension of road dust, which is enhanced at drier environmental conditions, is another major source of the airborne metals (Wang et al., 2005; Thorpe and Harrison, 2008; Chen et al., 2012). Therefore, EC is mostly influenced by the traffic intensity and atmospheric mixing height (Lin et al., 2009; Mues et al., 2017), which is probably the reason for its peak at morning (traffic)

and at night (low mixing height). In contrast, the ambient RH influences the road dust resuspension, yielding a different pattern in the metal concentration (highest during afternoon and lowest at night). This has been mentioned in the manuscript:

Page 13, lines 4 - 8

"The difference in the diurnal trends of metals and EC concentrations (i.e. EC peaks in the morning while metals peak in the afternoon) indicates a lesser influence of direct vehicular exhausts on the metals concentrations than the resuspended dust, which is generally driven by higher vehicular speeds (due to relatively lower traffic) and drier conditions in the afternoon (Pant and Harrison, 2013). Figure S6 in SI shows the diurnal pattern of ambient RH at the sampling site. The very high RH ($> 75\,\%$) substantially suppresses the dust resuspension and the resultant metals concentrations during nighttime."

7. Throughout the manuscript, the authors have used OP to describe DTT consumption rate per volume of air, whereas it is also often used to describe DTT consumption rate per mass of PM. I suggest defining it outright in the introduction. The per volume DTT consumption is often referred to as the "extrinsic" oxidative potential, or the oxidative capacity.

We agree with the reviewer here. We have replaced oxidative potential reported in our study as extrinsic DTT activity and denoted it as $OP_{ex}$ throughout the manuscript. We have further defined it outright on section 2.2 (where we have discussed about normalizing it with the volume of air) and made changes in the abstract

Page 1, lines 8 - 9

"We developed an online instrument for measuring the extrinsic oxidative potential (OP) of ambient particulate matter (PM) using the dithiothreitol (DTT) assay."

We have added the following sentence in section 2.2:

"This DTT consumption rate was normalized by the volume of sampled air and reported in the units of nmol $min^{-1}$ $m^{-3}$; referred hereafter as extrinsic DTT activity or $OP_{ex}$."

Minor technical comments:

 Page 2 line 9: times

We have modified the sentence as:

Page 2, line 9

"These assays are laborious, complex and require long analysis times (Dungchai et al., 2013)."

Page 2 line 10: what does it mean by "less controlled environment". Cell-free assays would be conducted under a more well-controlled environment than cellular assays.

What we meant here is that cell-free assays are easier to perform, but we agree with the reviewer that this sentence is confusing. Therefore, we have deleted the term "less controlled environment".

The revised sentence reads as:

Page 2, lines 10

"Cell free assays, on the other hand, are easier to perform and provide faster estimation of OP (Fang et al., 2015)."

Page 2 line 34: the word "data" is plural; remove "a"

Thank you for pointing out this error. We have made the suggested changes in the manuscript.

Page 2, lines 34

"Moreover, it requires relatively long sampling duration, i.e. at least six hours to collect enough volume of the concentrated slurry, making this approach unsuitable for obtaining highly time-resolved OP data for the ambient particles."

Page 5 line 14: "the" analytical part

Thank you for pointing out this error. We have made the suggested changes in the manuscript

Page 5, line 14

"While the analytical part of the instrument measured the DTT activity of a given PM suspension, the MC simultaneously collected a new PM suspension."

Page 5 line 30: "the" analytical part

Thank you for pointing out this error. We have made the suggested changes in the manuscript

Page 5, line 30

"The instrument performance was assessed by calibrating the analytical part (DTT activity determination) of the instrument using positive controls..."

Page 7 line 19: replace "reduces" with "is reduced" or "decreases"

We have modified the sentence as:

Page 7, line 19

"During air sampling, the volume of water inside the MC decreases due to evaporative loss."

Page 7 line 31: "The second concern"

Thank you for pointing out this error. We have made the suggested changes in the manuscript.

Page 7, line 31

"The second concern associated with the evaporation of water is the variable volume of the PM suspensions collected after different sampling runs depending on the ambient…"

Section 3.3: It may be more useful to report LOD in terms of oxidative capacity in nmol DTT/(min-m3 air)

We have reported the LOD in units of nmol/min, because it is consistent with earlier studies using automated (Fang et al., 2015) and online DTT activity measurement systems (Eiguren-Fernandez et al., 2017) and would help the readers to make an easy comparison with those systems.

LOD in the units of nmol/min/$m^3$ is operation-specific because it can be adjusted by changing the sampling duration in MC. For example, in an environment, where the PM is less redox active, more mass needs to be collected for the DTT activity analysis by increasing the sampling duration in MC. Therefore, we prefer to report the LOD in the same units (nmol/min) as used by other researchers.

Page 9 line 24: "the time series"

We have made the changes suggested by the reviewer

Page 9, line 24

"Figure 3 shows the time-series of the hourly ambient $PM_{2.5}$ DTT activity (blank-corrected) measured by the online instrument between May 31, 2017 and August 16, 2017."

Section 3.6.1: EC might be more a marker of diesel traffic

We agree with the reviewer's comment and have included this point in the manuscript.

Page 11, line 22

"EC can be assumed as a marker of exhaust emissions from diesel vehicles (Shah et al., 2004; Shirmohammadi et al., 2016)."

Page 11 line 24: awkward language: "getting elevated"

We have modified the sentence as:

Page 11, line 24

"It then subsides in the afternoon and remains constant till evening, before increasing again at night."

Page 12 line 33: there may be too many significant digits for Fe concentrations

We have modified the sentence as:

Page 12, line 33

"However, the concentration of Fe (3.7 - 6.9 ng/$m^3$) was significantly lower than reported in other studies"

Page 24 line 6: again, the word "data" is plural; replace "was" with "were"

Thank you for pointing out this error. We have made the suggested changes in the manuscript.

Page 24, line 6

"Figure 4. Diurnal profile of the ambient $PM_{2.5}$ DTT activity measured at the sampling site over (a) weekdays (n = 18 days), (b) weekends (n = 10 days). The data from May 31 to July 2, 2017 were used for plotting this profile. Error bars are the standard deviation (1σ) of the average DTT activity in that hour."

[revised manuscript text omitted]

---

## Author Comment (AC2) · 31 Aug 2018

**Response to Anonymous Referee #3**

Review of "Development and field-testing of an online instrument for measuring the real-time oxidative potential of ambient particulate matter based on dithiothreitol assay" by J. V. Puthussery et al.

This study describes the development of an automated, on-line system to measure the oxidative potential (OP) of particulate matter using the DTT assay. Measurements of OP and DDT activity have been increasing substantially in recent years with the acknowledgement that this may represent an indicator of the toxicity of PM. The system described in this manuscript represents the first automated measurement of DTT activity, and is therefore an important contribution to the atmospheric measurement community. Overall, the authors present a detailed, careful, and deliberate characterization of the system performance. The extensive ambient deployment of the system goes beyond most method papers. The writing is clear, the paper is well organized, and the figures are of high quality. I have several issues with the manuscript: I see some problems with the authors' attribution of the DTT activity to specific sources – it seems they are trying to explain their results based on what they expect, which does not necessarily agree with their actual observations. I also think that the different contributions of insoluble particles to the mist chamber samples is a bigger deal than the authors describe, and certainly contributes to the differences in OP between the offline and online measurements, but also to their troubles attributing the DTT activity to specific sources. I recommend the manuscript for publication after my comments have been addressed.

**Response**

We thank the reviewer for the insightful questions and comments. Our responses and corresponding changes made in the manuscript (highlighted in red) are given below.

**Specific Comments:**

1. I assume that this has been addressed in the references cited, but some discussion of the particle collection efficiency by the mist chamber is needed.

We agree with the reviewer's comment and have added some discussion in section 2.1. Note, previous studies have characterized the collection efficiency of MC only for the water-soluble components. For example, Hennigan et al. (2009) reported a collection efficiency of 95% with nitric acid aerosols at a flow rate of 21 LPM. King and Weber (2013) also calculated the collection efficiency of MC by simultaneously operating a particle into liquid sampler and measuring the sulfate collected by both systems. The sulfate collection efficiency in the MC was close to 100 % at flow rates greater than 25 LPM but was below 80% at flow rates less than 15 LPM. The air flow-rate of MC is directly related with the formation of jet spray of water, i.e. a high flow rate is required for a stronger jet spray, which could rinse the hydrophobic filter and

bring the particles into the MC suspension. Note, we operated the MC in our current study at a much higher flow rate (42 LPM) than used in previous works. Therefore, it is reasonable to assume that the collection efficiency of our MC is almost 100 % for at least the water-soluble components. However, we cannot comment on the collection efficiency for the water-insoluble components as no prior study has investigated it. Our study is probably the first to suggest the collection of water-insoluble components in MC.

**Page 4, line 2:**

"however few studies (Anderson et al., 2008; King and Weber, 2013) have also used these devices for collecting the water soluble fraction of the ambient particles. Hennigan et al. (2009) estimated the collection efficiency of MC as ~95% using nitric acid aerosols at a flow rate of 21 LPM. King and Weber (2013) also calculated the collection efficiency of MC by operating it simultaneously with a PILS and measuring the sulfate collected by both systems. The collection efficiency in the MC was close to 100 % at flow rates greater than 25 LPM."

2. The difference between the MC ROS and filter ROS is quite surprising (50% higher in the online system), and definitely indicates the contribution of insoluble components. Some discussion of Phillips and Smith (2017) should be included here. I think the major implication of this is that a direct comparison of the MC and filter samples is not so straightforward. The MC samples include a lot of insoluble particles which clearly contribute to the DTT activity (similar to the MeOH extracted samples), which is not the case for the water-soluble extracts. This muddles the comparison of the DTT activity to the other chemical components.

We agree with the reviewer that a higher activity obtained from the MC indicates the contribution from water-insoluble components in DTT activity. In fact, our results are consistent with previous studies which show that the insoluble particles have a significant influence on the DTT activity and should be accounted while measuring the OP of ambient PM (Daher et al., 2011; Gao et al., 2017). However, we don't believe that including the discussion of Phillips and Smith (2017) is relevant to our study. Phillips and Smith (2017) reports that the suspended insoluble fraction of PM, which is not removed even after filtering the methanol PM extracts can cause interference in the absorbance measurements and thus would overestimate the measurement of brown carbon. We agree this is a very important aspect to consider while developing the protocol for the measurement of atmospheric brown carbon; however, there are important differences to be considered while comparing their method with ours. First, they used a cuvette method on a Cary 60 UV-vis spectrophotometer instead of the liquid waveguide capillary cell (LWCC) used in our setup. Note, they themselves stated that such particle extinction might not be present to the same degree with the use of a narrow-bore LWCC as others have used (Phillips and Smith, 2017; page 1116, section 3.1.2). Second, they observed the interference in the spectra obtained only from the methanol extracts and not in the water extracts, even after various degrees of filtration, i.e. 0.22 µm and 0.45 µm syringe filtration of the waterextracts. Finally, it is important to note that in the DTT assay protocol, we are not measuring the absorbance caused by the aerosols, but rather from 2-nitro-5-thiobenzoic acid (TNB, the reaction product of DTT with DTNB), that too at a specific wavelength of 412 nm, in

comparison to the broad range of wavelengths (300-800 nm) used for brown carbon measurements. Our DTT assay protocol requires the reaction volume to be diluted by 50 times (from the original concentration of PM extract used in the reaction) before passing it through the LWCC for absorbance measurement. This is due to a relatively high concentration of DTT (100  $\mu$ M) used in the reaction vial which could saturate the absorbance spectra. Therefore, any interference from the insoluble aerosol particles, if present, in the absorbance measurement will be negligible.

As for the second part of the comment (comparison of MC with the filter measurement), we agree that a direct comparison of the MC results with chemical components is not straightforward. However, our objective was not to point out the specific water-soluble components that might be participating in the DTT consumption. Rather, we wanted to provide the insights into the emission sources contributing to the DTT activity. For example, WSOC and metals were considered as the tracers for SOA (Cheung et al., 2012; Verma et al., 2014) and vehicular emissions (including vehicle-induced road dust resuspension) (Wang et al., 2005; Hulskotte et al., 2007; Thorpe and Harrison, 2008; Chen et al., 2012), respectively. Considering this, we have modified our discussion in section 3.6.2, replacing the specific components with the possible emission sources, as follows:

Page 13, lines 9 - 11:

"A close similarity of the diurnal profile of the water-soluble metals with DTT activity suggests a significant contribution of both vehicular emissions (the morning peak) and resuspended dust (in the afternoon) to  $PM_{2.5}$  OPex at the sampling site..."

3. Pg. 9, line 18 - 20: I disagree with this statement. The collection of insoluble particles does not indicate that the MC performs better than conventional filter collection and extraction, but rather that the MC is subject to an artifact that may need to be accounted for.

Please note that this particular statement was made in the context of DTT activity measurement using MC. Numerous studies have shown that in addition to the soluble components (i.e. either water- or methanol-soluble) in the PM, the insoluble fraction that remains after extraction, also influences the DTT activity (Daher et al., 2011; Eiguren-Fernandez et al., 2017; Gao et al., 2017). There have been recent efforts to incorporate this insoluble fraction by using methods such as, performing the DTT reaction directly on the filter instead of extracting the particles with a solvent (Gao et al., 2017), or directly collecting the particles into liquid (e.g. liquid spot sampler) followed by its DTT activity determination (Eiguren-Fernandez et al., 2017). MCs have been traditionally used for collecting the water-soluble PM fraction (Anderson et al., 2008; King and Weber, 2013) and therefore our first step was to compare the online system results with conventional filter measurements. What we found during this analysis is that the online system (MC samples) results were consistently higher than the conventional filter extraction using DI water, and were closer to the values obtained by extracting the water-soluble components

(Anderson et al., 2008; King and Weber, 2013), it implies that the additional contribution in MC DTT activity results is from the water-insoluble components, which are not captured in the conventional filter extraction techniques using DI. We don't understand how we can say this is an artifact, when we know that these water-insoluble components are actually present in the ambient PM and do contribute to the DTT activity, as reported in previous studies (Daher et al., 2011; McWhinney et al., 2011; Eiguren-Fernandez et al., 2017; Gao et al., 2017). Rather, it should be considered as an advantageous feature of MC, which provides a more holistic assessment of the PM oxidative potential, as indicated from our results discussed on page 9, lines 12 - 20 of the revised manuscript. Therefore, we disagree with the reviewer's perspective on considering this as an artifact of MC.

4. Discussion on Pg. 2, lines 25 - 27: I agree there is great utility in an on-line measurement of oxidative potential, but this argument is rather weak. If Cu is really the most important species driving DTT activity, is there any evidence that Cu undergoes chemical changes during filter sampling?

We want to clarify here that although our results indicate that Cu is one of the important species influencing the DTT activity, our results cannot be generalized for other locations and environment. The range of Cu concentrations measured at our study location (4-23 ng/m3) was highest among three commonly reported redox active transition metals (i.e. Cu, Fe and Mn). Therefore, it probably contributes more substantially to the DTT activity at our location. However, there are other species like organic compounds (e.g. SOA), which are also known to significantly influence the DTT activity. For example, Verma et al. (2012) reported that HULIS fraction (free from Cu) of the PM in Atlanta accounted for 60% of the water-soluble DTT activity. Similarly, although Fe doesn't play a major role in the DTT oxidation, it can enhance the 'OH generation in DTT assay in the presence of organic compounds (Xiong et al., 2017; Yu et al., 2018). Therefore, OP of the ambient PM is due to a combined effect of various transition metals, organic compounds and probably other redox active chemical species. Even in our current study, the SOA appears to have a significant contribution to the DTT activity (Page 12, lines 24 - 25); however, we have not performed a detailed analysis to quantify the relative contributions of each of these species (which is beyond the scope of our current study, but we plan to do it in future investigations). Although we agree that there are currently no studies suggesting the chemical changes of Cu during filter sampling, there are several studies indicating the loss or alteration in the speciation of other redox-active components during sampling or storage. For example, Eatough et al. (2003) and Daher et al. (2011) have reported a significant loss of semi volatile organic species [known to contribute to the DTT activity (Verma et al., 2009)] in the conventional filter sampling techniques. Sampling artifacts in the measurement of ammonium, nitrate, chloride, and sulfate are commonly reported in conventional long duration (> 24 h) filter sampling, thus influencing composition of the collected PM (Yao et al., 2001; Pathak et al., 2004). The loss of these inorganic ions can change the acidity and subsequently the solubility of the transition metals, thereby indirectly alter the OP of the collected PM (Fang et al., 2017). Moreover, ambient Fe (II) collected over filters has been found to reduce the oxidized manganese (a DTT-active metal as shown in Charrier and Anastasio, 2012) present on the filter

(Majestic et al., 2007). Therefore, based on this literature, it is safe to assume that any artifact, either associated with sampling or storage of the filters, which affects the stability or chemical characteristic of any of the redox active species has the potential to influence the OP. And, our MC sampling technique would potentially reduce the impact of these artifacts on the OP measurement.

We have included these references to further strengthen our sentence on page 2, line 25 - 27:

"Generally, PM collected over filters might undergo chemical alteration during sampling, storage and extraction procedures, such as loss of semi-volatile organic (Daher et al., 2011) and inorganic (Yao et al., 2001; Pathak et al., 2004) compounds, and change in the oxidation state of metals, e.g. Mn (Majestic et al., 2007). Some of these components are known to directly contribute to OP [e.g. semi-volatile organic compounds (Verma et al., 2009) and Mn (Charrier and Anastasio, 2012)], while others (e.g. inorganic ions) can affect it indirectly by altering the solubility of redox-active metals (Fang et al., 2017b)."

5. Pg. 3, line 5: I do not understand this sentence?

A widely accepted hypothesis on the mechanism for how PM2.5 affects human health is associated with the reactive oxygen species (ROS). The ROS can be either formed endogenously after particle deposition in the human respiratory tract or it can be present directly on the particle itself (Knaapen et al., 2004; Venkatachari and Hopke, 2008). Majority of the existing studies on online OP measurement system measures the latter, i.e. the particle bound ROS (Venkatachari and Hopke, 2008; Wang et al., 2011; King and Weber, 2013; Wragg et al., 2016; Zhou et al., 2017) while adopting the dichlorofluorescein (DCFH) fluorescence probe to determine the ROS concentration. However, the in situ formation of free radicals after particle inhalation due to the presence of highly redox active species (such as transition metals, polycyclic aromatic hydrocarbons) is another major pathway for causing the PM induced oxidative stress in human body, which is probably more important than the particle bound ROS as suggested by Ayres et al., 2008. To the best of our knowledge, only two online systems have been developed till now, which measure the OP based on DTT assay, i.e. Sameenoi et al., 2012 and Eiguren-Fernandez et al., 2017. Therefore, we stated "However, most of these instruments measure the particle-bound ROS, which represents only a small part of the particles' OP", on page 3, line 5. We hope it is now clear to the reviewer.

6. Pg. 3, lines 17 - 20: this is somewhat misleading, since the ancillary components (syringe pumps, distribution valves, LWCC, spectrometer) will in total cost many thousands of dollars. My guess at the total system cost is \$12k - \$16k, and while this represents a lower cost than a method using the PILS or LSS, I don't think this qualifies as "low-cost".

Our intention here was to compare only the MC system to other existing sample collection systems (LSS or PILS) which are significantly costlier. But we agree with the reviewer that in combination with ancillary components, it doesn't qualify for the "low-cost" system. Therefore, we have omitted the "low cost" term from the manuscript and removed the cost comparison discussion with other online systems.

Page 3, lines 16 - 21

"Here, we discuss the development of an automated online instrument to measure the hourly averaged OP of ambient  $PM_{2.5}$  using DTT assay and its evaluation in the field conditions for over 50 days. A custom-built glass MC was used for collecting the ambient PM suspension, which is then transferred to an automated analytical system for DTT activity determination."

7. Section 2.2: give the manufacturer and purity of all chemicals and reagents.

We have added the manufacturer and purity information for all the chemicals.

8. Section 3.2: the strong RH-dependence on the remaining water volume is not consistent with other M.C. studies (e.g., Hennigan et al., 2018) – this may be due to inconsistencies between different mist chambers, but this difference is worth mentioning.

We agree with the reviewer that the strong dependence of the evaporative loss in MC on RH as seen in our study was not observed in previous studies. But, it is not due to inconsistencies between different mist chambers. There are two reasons for that: flow rate and the sampling duration, both of which were much higher in our system than those used in the previous studies. The flow rate in our system was 42 LPM compared to 21-28 LPM in previous works (Hennigan et al., 2009, 2018; King and Weber, 2013). Similarly, the duration of sampling in our system was 60 minutes compared to 5 minutes used in earlier studies. Therefore, the evaporative losses were minimal in those other studies but cannot be ignored in our case. Following the reviewer's suggestion, we have added a discussion on this in the manuscript:

Page 7, lines 19 - 22

"During air sampling, the volume of water inside the MC decreases due to evaporative loss. There are two concerns associated with this loss of water. First, if the water level drops below the capillary, the mist formation is stopped, and the filter will collect particles by dry sampling (i.e. without mist formation). We found that the rate of evaporation is largely governed by the ambient relative humidity (RH), which changes diurnally. Note, strong dependence of the evaporative loss in MC on RH as seen in our study was not observed in previous studies (Hennigan et al., 2009, 2018; King and Weber, 2013). This is probably due to significantly higher flow rate (42 LPM) and longer sampling duration (60 minutes) adopted in our study as compared to previous studies (flow rate = 21 - 28 LPM; sampling duration ~ 5 minutes). Therefore, the evaporative losses were minimal in those studies (Hennigan et al., 2009, 2018; King and Weber, 2013) but cannot be ignored in our case."

9. I really do not understand the results in Fig. S3 (and associated discussion in Pg. 8 lines 10 - 25). Why does a 50% change in the sample volume (either + or -) not result in a 50% change in the DTT activity measurement?

First, we would like to clarify here that a 50 % change in the sample volume does not result in a 50 % change in the reaction volume. Reaction volume consist of 1.75 mL sample + 0.5 mL buffer + 0.25 mL DTT (total reaction volume = 2.5 mL). Therefore, a 50 % change in the sample volume will result into a 35 % change in the reaction volume. Second, a change in the reaction volume is also accompanied by a corresponding change in initial DTT concentration in the reaction vial, which influences the self-oxidation rate of DTT (Sauvain and Rossi, 2016). Finally, different components in ambient PM can have different response functions with the DTT oxidation as shown by Charrier and Anastasio, 2012, and thus can result into a varying degree of bias in the DTT activity measurements caused by a change in the reaction volume.

To further understand this, consider the following example. Let's assume we have 1.75 mL of PM extract of concentration 10 µg/mL (thus total PM mass =  $10x1.75 = 17.5 \mu g$ ). If we dilute this extract to 2.625 mL (i.e. dilute it by 50 %), and then add it to the reaction vial for DTT assay, the total reaction volume will increase to 3.375 mL (a 35 % increase in the reaction volume). But, the initial DTT concentration in the reaction volume will reduce from 100 µM to 74 µM (a 25 % change), which will also affect the auto-oxidation rate of DTT. Note, the total amount of PM available to catalyze the oxidation of DTT is still same, i.e. 17.5 µg as in the original reaction volume (i.e. 2.5 mL). Given, different components in PM (e.g. Cu, Mn, quinones etc.) have different response curve with the DTT oxidation rate, it is difficult to predict the net impact on the measured DTT activity. Therefore, we conducted this experiment using four different PM samples (i.e. to account for the variations in chemical composition) and empirically derived the effect of change in the sample volume on the measurement bias for DTT activity. As mentioned in the manuscript, maximum bias in the DTT activity for a 20 % variation in the sample volume was less than 6 % (average  $3 \pm 3$  %); therefore, this small bias was neglected for the purpose of this study.

10. Pg. 12, line 10 - 11: I don't agree with this logic – the authors state that traffic at their site after 10pm is almost nonexistent, which would suggest some of the EC is aged.

Here we would like to again clarify that the "night" time filter sample was collected from 7 PM in the evening to 7 AM the following morning. Therefore, we expect that most of the EC collected on the filter will be from the evening traffic (7 - 10 PM), which is still fresh. Moreover, yes, the traffic at the site after 10 pm is almost nonexistent, but there will be no photochemical reactions at night. Even the ozone concentrations are at their minimum from 10 PM - 7 AM (Figure S5). Therefore, we don't believe that EC will be significantly aged at this site. Finally, given the site is adjacent to a busy roadway, it is reasonable to assume that most of the EC (if not all) is fresh and unoxidized.

11. Pg. 12, line 23 - 24: this does not seem consistent with the data in Figure 5. The WSOC diurnal profile is flat throughout the day. Most (if not all) of the WSOC during summer at this location should be secondary – how then do the authors attribute the afternoon increase in DTT to SOA?

To explain why we attribute this increase in the afternoon DTT activity to SOA, we would like to first explain the EC diurnal trend. The EC profile shown in Figure 5 was similar to those reported by several other studies (Mues et al., 2017; Sharma et al., 2017; Singh et al., 2018) where they have attributed the low EC concentration during daytime to an increase in the mixing layer height. Now, if all of the OC at our site is only from the background SOA as suggested by the reviewer, then the OC and WSOC profile should follow the same diurnal profile as EC, i.e. low concentration during afternoon but higher in early morning and at nighttime. However, that's not the case, rather both OC and WSOC remain flat. Therefore, there must be some additional OC contribution during the afternoon period. To further test this, we plotted the ratio of OC/EC and WSOC/EC in Figure 5 (assuming EC as a conservative species emitted only from primary sources such as diesel vehicles), which follows a diurnal trend, i.e. high in the afternoon and low at night. Please note that OC/EC ratio has been used by several other researchers to predict the contribution of SOA to the total OC (Turpin and Huntzicker, 1995; Castro et al., 1999; Cabada et al., 2004; Pio et al., 2011) and we have followed the same approach here. Thus, the increase in the OC/EC or WSOC/EC ratio during the afternoon is a clear indicator for an additional source of OC, which we hypothesized to be from the fresh SOA formation at the site in afternoon.

This has been explained in the manuscript as well:

Pages 12, lines 15 - 24

"As depicted in Fig.5, neither OC nor WSOC show any diurnal pattern. Figure 5 also shows OC/EC ratio, which peaked in the afternoon. Considering a higher mixing height and relatively lower traffic in the afternoon than morning, an elevated OC/EC ratio indicates an additional OC contribution, which compensates its decrease from reduced vehicular emissions and enhanced atmospheric mixing. We attribute this additional OC to the secondary particle formation via photochemical reactions, which keeps the OC concentration almost constant throughout the day. Figure S5 in SI shows a diurnal profile of ozone measured at Bondville (EPA site). The ozone concentration peaked from 11:00 AM to 6:00 PM indicating secondary formation of particles in the afternoon period. To further confirm the contribution from SOA to OC, WSOC/EC ratio was also plotted (Fig.5), which ranged from 6.6 (morning) to 11.3 (evening) and followed a similar diurnal profile as ozone or OC/EC ratio. Thus, the broad peak in DTT activity during afternoon and evening periods could partly be caused by the redox-active SOA components."

12. Pg. 13, lines 4 - 11: I disagree with this interpretation of the data. I do not think that Figures 4 and 6 show a strong contribution of Cu to the measured OP. See prior comments, but it's important to acknowledge that Figures 4 and 6 cannot be directly compared due to the contribution of insoluble particles to the Fig. 4 measurements.

By comparing the data in Figures 4 and 6, we want to show that the diurnal trend of Cu measured on the water-soluble PM fraction of the time segregated filter samples and the DTT activity measured by the online system is qualitatively similar (i.e. high levels during daytime while low at night). Our purpose of investigating this association is not to imply the direct contribution of water-soluble Cu to the measured DTT activity but indicate towards the possible emission sources, i.e. vehicular exhausts and road-dust resuspension contributing to the DTT activity. Therefore, water-soluble Cu shown Figure 6 and its comparison with Figure 4 should be interpreted as the markers for the emission sources which could have a substantial impact on the DTT activity of ambient PM. Please note that we do acknowledge the reviewer's comment that the Figures 4 and 6 cannot be directly compared due to the contribution of insoluble particles to the measurements shown in Figure 4. Therefore, considering his/her comment, we have modified the associated text in the abstract, conclusion and in the results and discussion section:

Page 1, lines 24 - 26

"Based on this comparison, we attributed the daytime OP of ambient PM2.5 to the vehicular (both exhaust and non-exhaust) emissions and resuspended dust, whereas secondary photochemical transformation of primary emissions appear to enhance the OP of PM during the afternoon and evening period."

Page 13, lines 9 – 11

"A close similarity of the diurnal profile of the water-soluble metals with DTT activity suggests a significant contribution of both vehicular emissions (the morning peak) and resuspended dust (in the afternoon) to  $PM_{2.5}$  OPex at the sampling site."

Page 13, lines 32 - 33 and page 14, lines 1 - 2

"By comparison of the DTT activity with various chemical components, i.e. OC, EC, WSOC, Cu, Fe and Mn, the morning peak in DTT activity profile was attributed to the vehicular sources (exhausts and non-exhausts), whereas both secondary formation (i.e. SOA) and resuspended dust seem to contribute to the afternoon peak."

**Technical Corrections:**

Pg. 1, line 9: "using the dithiothreitol..."

The sentence has been modified as:

Page 1, lines 8 - 9

"We developed an online instrument for measuring the extrinsic oxidative potential (OP) of ambient particulate matter (PM) using the dithiothreitol (DTT) assay."

Pg. 1, line 17: date format should be 4 July

The sentence has been modified as:

Page 1, lines 17 - 18

"However, a four-fold increase in the hourly averaged activity was observed on the night of 4 July (Independence Day fireworks display) ..."

Pg. 2, line 17: delete comma before 'PM'

Comma has been deleted

Page 2, line 17

"Previous studies have suggested that OP of the ambient particles is affected by various factors such as PM composition, size, and source (Li et al., 2003; Steenhof et al., 2011; Janssen et al., 2014; Tuet et al., 2016; Fang et al., 2017a)."

Pg. 4, line 28: try to avoid beginning a sentence with a number

The sentence has been modified as:

Page 4, line 28

"DTT [0.25 mL, 1 mM, final concentration in the reaction vial = 100  $\mu$ M (Sigma Aldrich, St. Louis, MO, USA; > 99% purity)] and potassium phosphate buffer [0.5 mL, pH 7.4, 0.5  $\mu$ M (Sigma Aldrich, St. Louis, MO, USA; > 99% purity)] were then added to the RV..."

Figure 3: I believe the incorrect dates are given in the caption for the time the instrument was not operated.

We thank the reviewer for pointing out this error. We have corrected the dates in the revised manuscript.

"Figure 3. Time-series plot of the DTT activity from May 31,2017 to Aug 16, 2017. The shaded portions in the graph are weekend DTT activity measurements. The instrument was not operated between July 12 and Aug 3, 2017 (shown as a break in the X-axis)."

---

## Referee Report (RR1)

Comments to "Development and field-testing of an online instrument for measuring the real-time oxidative potential of ambient particulate matter based on dithiothreitol assay" by Joseph V. Puthussery et al.

The revised manuscript improved a lot. I only have one minor comment to the interpretation of the difference between the online and offline DTT activity in Figure 2. In the section 3.4, the authors claimed that 'The higher $OP_{ex}$ measured directly on the filters was attributed to the contribution from water-insoluble PM fraction remained on the filters, which is not fully extracted even by methanol.' But I did not see any convincing evidence from the manuscript to support this explanation. There are maybe other plausible reasons that may explain such a difference, e.g. dissociation or decomposition of PM-bound redox active substances by chemical aging and sonication. The chemical substances that have high DTT activity are generally reactive and not so stable. Therefore, discussions about this aspect will be helpful and beneficial to readers. Overall, I prefer to recommend the publication of this manuscript in AMT finally.

---

## Author Response (AR2)

Response to Referee 4 comments:

Comments to "Development and field-testing of an online instrument for measuring the realtime oxidative potential of ambient particulate matter based on dithiothreitol assay" by Joseph V. Puthussery et al.

The revised manuscript improved a lot. I only have one minor comment to the interpretation of the difference between the online and offline DTT activity in Figure 2. In the section 3.4, the authors claimed that 'The higher $OP_{ex}$ measured directly on the filters was attributed to the contribution from water-insoluble PM fraction remained on the filters, which is not fully extracted even by methanol.' But I did not see any convincing evidence from the manuscript to support this explanation. There are maybe other plausible reasons that may explain such a difference, e.g. dissociation or decomposition of PM-bound redox active substances by chemical aging and sonication. The chemical substances that have high DTT activity are generally reactive and not so stable. Therefore, discussions about this aspect will be helpful and beneficial to readers. Overall, I prefer to recommend the publication of this manuscript in AMT finally.

We thank the reviewer for his/her recommendation on the acceptance of the paper. As per the reviewer's suggestion, we have included some discussion (section 3.4) pertaining to the possibility of artifacts/chemical alterations of the PM collected on the filters during storage and/or sonication, which probably contributes to the lowered DTT activity than obtained by the online system.

"The lower DTT activity measured by offline filter extraction method in our study could also be due to the loss of redox active semi-volatile species and other chemical alterations in the collected PM during storage (Yao et al., 2001; Pathak et al., 2004; Daher et al., 2011), and extraction of the filters by sonication (Roper et al., 2015). 
[revised manuscript text omitted]